# Prioritization of potential causative genes for schizophrenia in placenta

Gianluca Ursini [1,2] ✉, Pasquale Di Carlo[1,3], Sreya Mukherjee[1], Qiang Chen[1], Shizhong Han [1], Jiyoung Kim[1], Maya Deyssenroth[4], Carmen J. Marsit [5], Jia Chen[4], Ke Hao [6], Giovanna Punzi [1,2] & Daniel R. Weinberger [1,2,7,8,9] ✉

Our earlier work has shown that genomic risk for schizophrenia converges with early life complications in affecting risk for the disorder and sex-biased neurodevelopmental trajectories. Here, we identify specific genes and potential mechanisms that, in placenta, may mediate such outcomes. We performed TWAS in healthy term placentae ($N = 147$) to derive candidate placental causal genes that we confirmed with SMR; to search for placenta and schizophrenia-specific associations, we performed an analogous analysis in fetal brain ($N = 166$) and additional placenta TWAS for other disorders/traits. The analyses in the whole sample and stratifying by sex ultimately highlight 139 placenta and schizophrenia-specific risk genes, many being sex-biased; the candidate molecular mechanisms converge on the nutrient-sensing capabilities of placenta and trophoblast invasiveness. These genes also implicate the Coronavirus-pathogenesis pathway and showed increased expression in placentae from a small sample of SARS-CoV-2-positive pregnancies. Investigating placental risk genes for schizophrenia and candidate mechanisms may lead to opportunities for prevention that would not be suggested by study of the brain alone.

The placenta, the first organ to form in mammals, is critically involved in early stages of development, establishing a maternal-embryo/fetal interface that supplies the bioenergetic needs of the conceptus. In addition to providing nutrients and oxygen support, the placenta disposes waste products, prevents perilous immune responses, and protects the embryo/fetus from infections[1]. All of these functions are essential for the development of the brain[2]. Indeed, conditions associated with a disruption of such functions are also associated with an increased risk for neuropsychiatric disorders, such as schizophrenia[2,3].

Schizophrenia is a complex disorder, affecting approximately 1% of the world's population[4], with a higher incidence in male individuals, compared with females[5]. While the onset is usually dated to late adolescence or early adulthood, the trajectories of risk start early in life, consistent with the neurodevelopmental origins of the disorder[6]. Such trajectories are likely affected by genetic susceptibility[7]. The heritability of the disorder is estimated between 50 and 80%[8], and recent Genome Wide Association Studies (GWAS)[9,10] have identified hundreds of genetic loci associated with the disease, allowing for the calculation of aggregate metrics of genomic risk[11,12], which account for around

[1]Lieber Institute for Brain Development, Johns Hopkins University Medical Campus, Baltimore, MD, USA. [2]Department of Psychiatry and Behavioral Sciences, Johns Hopkins University School of Medicine, Baltimore, MD, USA. [3]Group of Psychiatric Neuroscience, Department of Basic Medical Sciences, Neuroscience and Sense Organs, University of Bari Aldo Moro, Bari, Italy. [4]Departments of Environmental Medicine and Public Health, Icahn School of Public Health at Mount Sinai, New York, NY, USA. [5]Departments of Environmental Health and Epidemiology, Rollins School of Public Health, Emory University, Atlanta, GA, USA. [6]Department of Genetics and Genomic Sciences, Icahn School of Medicine at Mount Sinai, New York, NY, USA. [7]McKusick-Nathans Institute, Department of Genetic Medicine, Johns Hopkins School of Medicine, Baltimore, MD, USA. [8]The Solomon H. Snyder Department of Neuroscience, Johns Hopkins School of Medicine, Baltimore, MD, USA. [9]Department of Neurology, Johns Hopkins School of Medicine, Baltimore, MD, USA. ✉e-mail: gianluca.ursini@libd.org; drweinberger@libd.org

7.3% of the liability to the disorder[10]. However, we still do not adequately understand the mechanisms that link genomic risk to illness and, although genetic studies of the disorder have been around for more than half a century, no incisive prevention or treatment strategy has been developed based on the knowledge derived by them.

The commonly shared view on schizophrenia etiopathogenesis is that genetic variants associated with risk play a role directly in the brain, by inducing changes in the expression of genes (e.g., schizophrenia risk-associated genes) that might disrupt brain development and function. This view is supported by experimental evidence from animal models[13–15] and in vitro studies[16–18], and from research in humans[15,19,20], showing the effect of these genes on selected brain phenotypes. Recently, the integration of GWAS with expression mapping studies has led to the development of computational approaches[21–26] that allow prediction of gene expression in specific tissues for each individual in GWAS case-control samples. In this way, statistical associations can be estimated between predicted gene expression and a trait, in samples much larger than those usually available in transcriptomic studies. These approaches include Transcriptome Wide Association Studies (TWAS)[22,26], which have the advantage of aggregating the effects of multiple SNPs onto specific genes, reducing multiple comparisons, and increasing power for association testing[21,22], and Summary data based Mendelian Randomization (SMR), a complementary method for causative gene inference that helps to distinguish linkage from pleiotropy[25,26]. These approaches have allowed the identification and prioritization of candidate causal genes for schizophrenia, in brain areas whose development is thought to be altered in the disorder, such as cerebral cortex[26–28], hippocampus[27], and caudate[29]. While most of these studies are based on datasets from adult individuals, recent transcriptomic and epigenetic studies also indicate that some of the putative schizophrenia risk genes are dynamically regulated in brain in early development[30–32]. Indeed, some candidate causal genes for schizophrenia have also been highlighted by TWAS and SMR studies performed in fetal brain[10,33].

Because complications during prenatal and perinatal life (Early Life Complications, ELCs), represent the most common environmental risk factor associated with schizophrenia in offspring[3,34], we have previously investigated the possibility of a convergence between ELCs and genomic risk for schizophrenia[35,36]. We found that genomic risk for schizophrenia interacts with ELCs so the liability for the disorder associated with genomic risk is up to five times higher in the presence of ELCs, compared to when ELCs are absent[35]. We also found that genes in schizophrenia risk loci are highly expressed in placenta and, specifically, in placentae from complicated pregnancies; moreover, they are more highly expressed in placentae from male compared with female offspring[35]. By computing an index of placental genomic risk for schizophrenia based on gene expression in placenta, we found that placental gene expression appears to mediate the interaction between genomic risk and ELCs on case-control status[35]. The same index of placental genomic risk was associated with early neurodevelopmental trajectories of risk, as shown by its association with neonatal brain volume, and developmental scores at one year of age, particularly in male individuals[36]. One subsequent study[37] that used a less detailed inventory for the assessment of ELCs and that did not survey many of the ELCs associated with schizophrenia[38,39] did not observe a statistical interaction between genomic risk and ELCs[37]. In contrast, the principal findings from our two studies[35,36] have been largely replicated in a recent study from Norway[40], which detected an interaction between placental genomic risk for schizophrenia and serious ELCs associated with birth asphyxia. These results have triggered broader perspectives in gene-by-environment research related to psychiatric illness[41] and strongly suggest that some of the schizophrenia-associated variants may converge on mechanisms of risk linked with ELCs and placenta biology, leading to altered trajectories of development that start very early in life, are particularly relevant in males, and are potentially reversible.

Here, we identify and investigate specific genes that, in placenta, mediate these potential mechanisms. We use transcriptome data from healthy placentae at term to identify and prioritize placental genes, and their isoforms, with a potential causal role for schizophrenia. To this purpose, we first perform a TWAS to derive a list of candidate placental genes that we then confirm based on SMR; to search for placenta-specific associations, we also perform an analogous analysis in fetal brain. Because our prior data[35,36] pointed towards sex-biased placental processes relevant for brain development, we further perform analyses separately by sex. We extend these results with a colocalization analysis and with independent replication. Finally, we perform placental TWAS for other disorders and traits, which further supports how the disturbances in the placental transcriptome are relatively specific to schizophrenia (workflow and prioritization strategy in Supplementary Fig. 1). While the results do not detract from the importance of gene expression in brain for schizophrenia risk, they reveal a wider stage including placenta.

## Results
### The placental genes associated with schizophrenia imply nutrient sensing, trophoblast growth, and invasiveness
Fastq files from 150 placenta samples previously selected as having good quality RNA sequencing and genotyping data[42] were downloaded from the NCBI SRA public database (SRP095910) and re-aligned to the Genome Reference Consortium Human Build 38 (GRCh38). The transcriptomes of these placentae were characterized by measurement of: (i) gene-level expression and (ii) transcript-level quantification, which relies on existing gene transcripts annotation[31], resulting in 58,037 genes and 198,093 transcripts, which became respectively 21,115 and 53,637 after filtering lowly expressed features. Further quality check and outlier detection (see Methods for details) led to a final sample of 146 placentae (73 females) for the gene level and 147 placentae (73 females) for the transcript-level data. Supplementary Data 1 summarizes the sample demographics. Features expression was then adjusted accounting for ten genomic principal components, mitochondrial RNA rate, ribosomal RNA rate, gene assignment rate, and RNA sequencing batch. We performed fusion TWAS[22] to identify genes expressed in placenta with a potential causal role in the etiology of schizophrenia, using GWAS summary statistics from the latest release from the Psychiatric Genomic Consortium[10], and both adjusted gene-expression (g-TWAS) and transcript-expression (tx-TWAS) as reference panels (see Methods for details). Linkage disequilibrium (LD) was considered using the European ancestry samples of the 1000 Genome phase III aligned to the GRCh38 as a reference panel. We here report results excluding the major histocompatibility complex (MHC) genomic region, because the high LD in this locus may enhance false discovery[43].

We detect 8558 genes and 18,582 transcripts (corresponding to 8523 unique genes) with heritable cis-regulated expression in placenta. The schizophrenia g-TWAS identifies 100 placental genes significantly associated with schizophrenia (Bonferroni corrected $p$-value: $p_{corr} \leq 0.05$; Fig. 1a; Supplementary Data 2), spanning 83 loci; 11 of these 100 placental schizophrenia risk genes lay in 8 loci that are not GWAS-significant (Supplementary Data 3). We also performed a tx-TWAS to search for specific placental gene isoforms associated with schizophrenia, whose expression is not related to gene-level summarized expression. The transcript level analysis (tx-TWAS) detects 235 transcripts significantly associated with schizophrenia ($p_{corr} \leq 0.05$), corresponding to 187 unique genes spanning 137 loci (Fig. 1b; Supplementary Data 4); 12 of these 235 transcripts lay in 9 loci that are not GWAS-significant (Supplementary Data 5). Overall, the two analyses highlight 262 unique genes with predicted expression in placenta associated with schizophrenia at gene and/or

transcript level. These 262 unique candidate placental schizophrenia risk genes are within 169 loci (Supplementary Data 6), and twenty are in 16 loci that are not GWAS-significant (Supplementary Data 7). Seven genes (*MARK3*, *RPS17*, *SPG7*, *SRA1*, *KANSL1*, *KIAA0319L*, *SNORD3B-1*) contained placental transcripts whose individual predicted expressions were associated with schizophrenia with opposite signs (i.e., higher expression of one transcript and lower expression of the other), suggesting a link between genomic risk for schizophrenia and splicing events in the placental transcriptome (Supplementary Data 6).

For orthogonal confirmation, we conducted SMR, a complementary method that tests for pleiotropic associations in the cis-window with an accompanying HEIDI test to distinguish linkage from pleiotropy[25]. Of the 262 placental genes prioritized by g- or tx-TWAS, 193 (73.66%) were further supported by SMR with concordant association with schizophrenia ($P_{SMR} < 0.05$, and with $P_{HEIDI} > 0.01$ as in previous work[44]) (Supplementary Data 8–10). The relationship between genomic risk for schizophrenia and splicing events in the placental transcriptome was supported by the SMR analysis for five out of the seven genes (*MARK3*, *SPG7*, *SRA1*, *KANSL1*, *KIAA0319L*) with placental transcripts associated with schizophrenia with opposite sign (Supplementary Data 10).

The placental genes and transcripts associated with schizophrenia risk were enriched for pathways associated with placental growth, protein synthesis, nutrients sensing, and trophoblast invasiveness. In particular, the gene expression changes associated with schizophrenia risk in placenta indicated EIF2 signaling pathways, inhibition of mTOR signaling, activation of the Estrogen Receptor, and Insulin secretion

and receptor signaling. Of note, we also detected activation of the Coronavirus pathogenesis pathway (Fig. 1c), enrichment for pathways involved in the pathogenesis of influenza (Supplementary Data 11), and an enrichment for annotations linked with cancer, consistent with the biological similarity between placental invasion and malignancy[45,46] (Supplementary Data 12). The upstream regulators enriched for the placental schizophrenia risk genes included UQCC3, a mitochondrial factor with a crucial role in bioenergetic reprogramming linked with hypoxia[47], serotonin—known to mediate the effect of maternal inflammation on offspring neurodevelopment[48]—, and PCB53[49] (also known as 2,4,5,2′,4′,5′-hexachlorobiphenyl), a chemical suspected to alter brain development (Fig. 1d; Supplementary Data 13).

## Sex-specific mechanisms influence the placental transcriptome and risk for schizophrenia

Because of previous evidence of sex-biased placental expression of genes in schizophrenia risk loci[35] potentially relevant in affecting neurodevelopmental outcomes[36], we hypothesized that placentae from female and male offspring would show some specific associations between gene expression and schizophrenia risk. We, therefore, performed a placental TWAS stratified by sex. The analysis in female placentae (N = 73; see Supplementary Data 1 for details) highlighted 107 placental genes (Fig. 2a; Supplementary Data 14) and 159 transcripts (137 unique genes, Fig. 2b; Supplementary Data 15) associated with schizophrenia, so that 225 unique genes were associated with schizophrenia at gene and/or transcript level in female placenta (Supplementary Data 16). These 225 genes span 146 loci, 12 of which are outside GWAS-significant loci; 124 of these genes did not emerge as

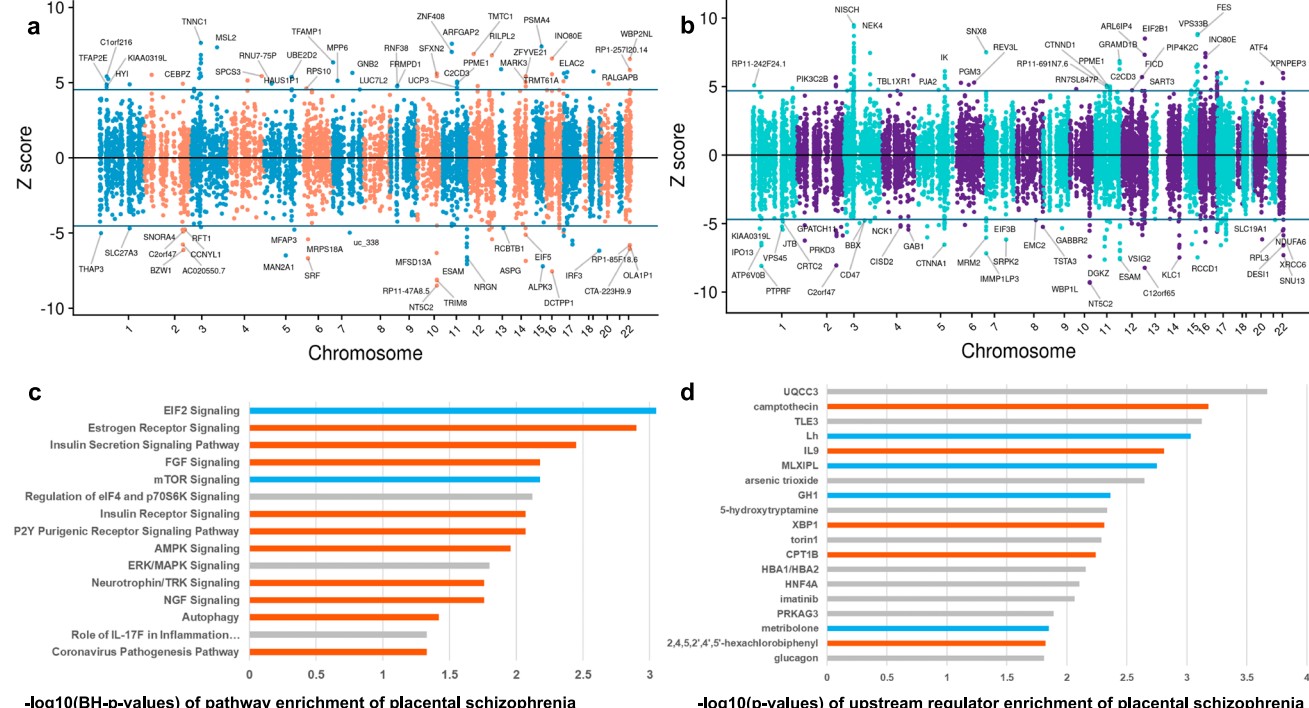

**Fig. 1 | Schizophrenia TWAS in placenta.** Results from TWAS prioritize genes (**a**) and transcripts (**b**) whose cis-regulated expression in placenta (N = 146 for the gene-level TWAS; N = 147 for the transcript-level TWAS) is associated with disease. Plots show conditionally-independent TWAS prioritized genes (**a**) and transcripts (**b**). The sign of TWAS z-scores (y-axis) indicates predicted direction of effect. Genes with predicted expression significantly up- or down-regulated in placenta are respectively above and below the horizontal lines corresponding to the TWAS level of significance after Bonferroni correction for multiple comparisons. 100 genes and

235 transcripts are significantly associated with schizophrenia (corresponding to 262 unique genes). Pathway (**c**, top 12 and selected) and upstream regulators (**d**, selected) enriched for the schizophrenia risk genes identified with the TWAS (N of genes = 262): bars depict negative logarithm of the *P* values (orange bars: activation, turquoise: inhibition, gray: absence of predicted directionality; negative logarithm of Benjamini−Hochberg corrected p-values are shown in (**c**), of uncorrected *p* value in (**d**) from right-tailed Fisher's Exact Test.

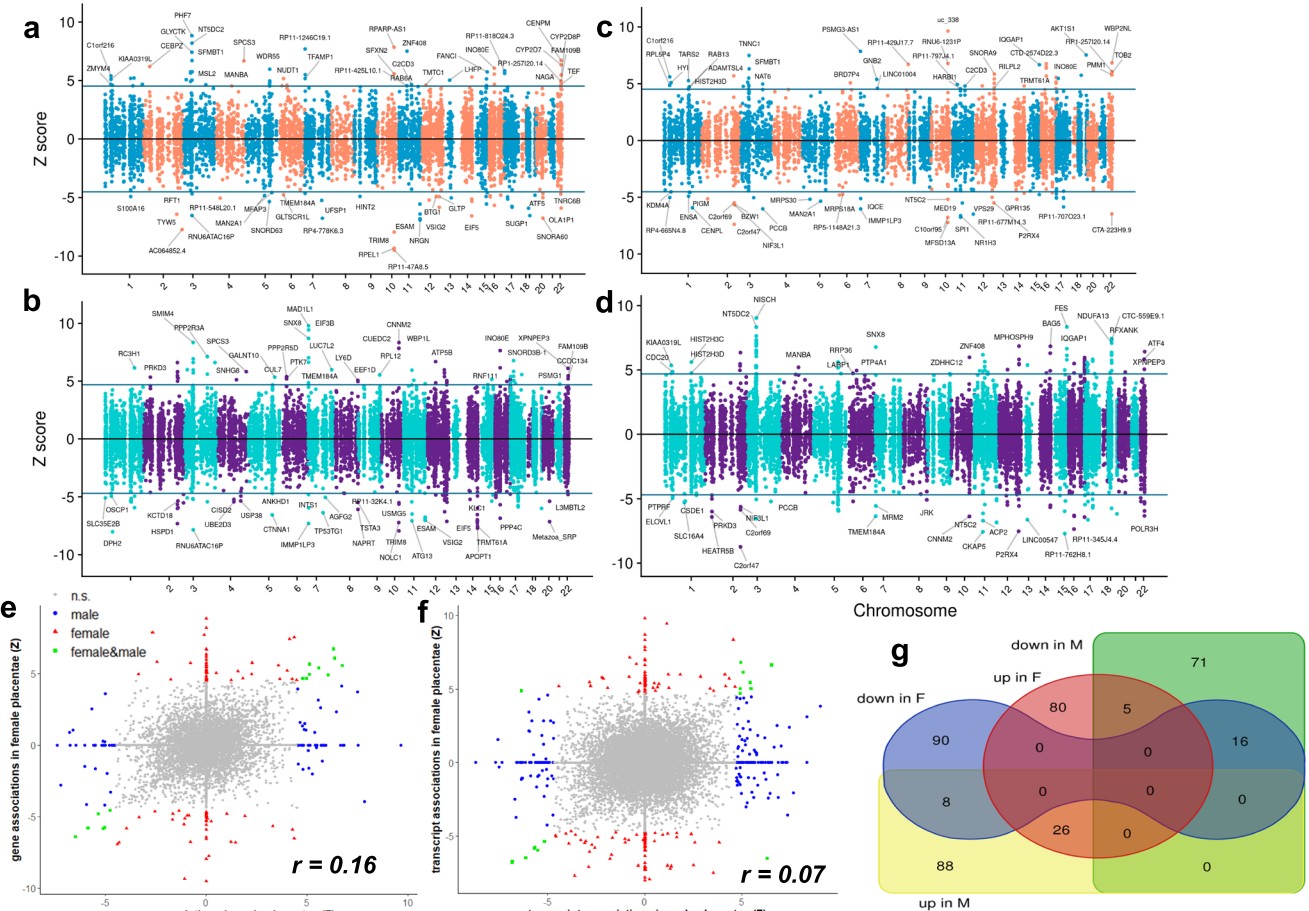

**Fig. 2 | Sex bias in placental TWAS for schizophrenia.** TWAS identifies genes and transcripts whose cis-regulated expression in female (**a**, **b**) and male (**c**, **d**) placentae is associated with schizophrenia ($N = 73$ and 73 for the gene-level TWAS in female and male placentae; $N = 73$ and 74 for the transcript-level TWAS in female and male placentae respectively). Plots show conditionally-independent TWAS prioritized genes. The sign of TWAS z-scores (y-axis) indicates predicted direction of effect. The horizontal lines indicate the TWAS level of significance after Bonferroni correction for multiple comparisons. Scatterplots of the correlation between the Z scores from the TWAS analyses in male (x-axis) and female (y-axis) placentae, at gene (**e**) and transcript (**f**) level. Highlighted are genes and transcripts that are TWAS-significant only in males (blue), only in females (red), and in both males and females (green). **g** Venn diagram of the overlap between the TWAS significant genes whose predicted expression, up-regulated in female placenta (up in F, red oval area), down-regulated in female placenta (down in F, blue area with two ovals), up-regulated in male placenta (up in M, yellow rectangular area), down-regulated in male placenta (down in M, green rectangular area), is associated with schizophrenia.

TWAS-significant in the analysis of the whole sample. The analysis in male placentae ($N = 74$) highlighted 91 placental genes (Fig. 2c; Supplementary Data 17) and 188 transcripts (150 unique genes, Fig. 2d; Supplementary Data 18) associated with schizophrenia, so that 215 unique genes were associated with schizophrenia at gene and/or transcript level in male placenta (Supplementary Data 19), spanning 140 loci, 11 of which are not GWAS-significant; 130 of these 215 genes did not emerge as TWAS-significant in the whole sample. One male-specific association highlighted *ATP5MGP8*, a pseudogene whose strongest placental eQTL was the functional SNP rs6265 (Val66Met) in the *BDNF* locus[50]. Another male-specific association pointed to the DiGeorge syndrome critical region 8 gene (*DGCR8*), which is crucial for regulating microRNAs of the C19-MC microRNA cluster, expressed in the placenta, and regulated by imprinting;[51–53] of note, haploinsufficiency of *DGCR8* has been associated with altered brain development[54]. Four genes in females (*MAD1L1, Metazoa_SRP, CNNM2, RPS17*) and eight genes in males (*BAP1, ACY1, KANSL1, CHMP1A, CTNNA1, EIF5, CLP1, SNORD3B-1*) contained placental transcripts whose individual predicted expression was associated with schizophrenia with opposite directionality, suggesting that the link between genomic risk for schizophrenia and splicing events in the placental transcriptome is not

biased by sex-related differences (Supplementary Data 16,19). SMR validated respectively 201/225 (89.33%) and 175/215 (81.40%) of the unique placental genes associated with schizophrenia in female and male placentae, respectively (Supplementary Data 20,21). Overall, the analyses in the whole sample, in male and female placentae, detected 502 placental genes associated with schizophrenia at gene and/or transcript level (Supplementary Fig. 1).

Interestingly, a comparison of the individual TWAS results in the two sexes revealed only 42 genes with a consistent TWAS-significant ($p_{corr} < 0.05$) association at gene or transcript level (10.91%) in both males and females, and 13 genes (including *EPN2, CORO1A, ATP5B, TCF25, MARS, KIAAO319L, ADAMTSL4*) associated with schizophrenia with opposite sign in male and female placentae (Fig. 2e–g, Supplementary Data 22). When using a less stringent statistical threshold (FDR < 0.05), the overlap between TWAS-significant genes in males and females was still limited (413/2173, 19%), as was the correlation between the Z scores of the two analyses ($r = 0.1629$ and $0.0718$ at gene and transcript level respectively).

Overall, the analyses stratified by sex suggest that the placental transcriptome includes genes that play a potential causal role in schizophrenia, through mechanisms that may be different in males and

females. To gain further insight about sex-related differences explaining the associations in males and females, we performed weighted genome coexpression network analysis (WGCNA)[55]. WGCNA allows the detection of pathways of coexpression of groups of genes, i.e., modules, which may be linked with specific biological functions, specific regulatory machinery, risk gene sets, and cell types. We performed separate WGCNA in the samples of female ($N = 73$) and male placentae ($N = 74$) using transcript-level expression data, after removing highly correlated transcripts of the same genes. We detected 64 modules of coexpressed gene isoforms across both samples (Supplementary Data 23, 24). We analyzed the overlap of gene memberships between modules of the two sexes, in order to detect commonalities and differences in the architecture of gene coexpression in male and female placentae networks (Supplementary Data 25, 26). Based on the significance of the overlap of gene compositions, we distinguished overlapping, female-specific, and male-specific placental modules (Supplementary Data 27), and we explored the enrichment for biological pathways and functions of the modules containing the placental TWAS-significant genes associated with schizophrenia (set1: whole sample TWAS, set2: female TWAS, set 3: male TWAS). Of note, only three modules in females and two modules in males significantly overlap, indicating a sex-related bias in placental gene coexpression (Supplementary Fig. 2, Supplementary Data 25, 26).

Among the overlapping modules, we identified one mTOR/EIF2 signaling module, which contained genes from all the sets of the placental schizophrenia TWASs (19 from set1: whole sample TWAS, 15 from set2: female TWAS, 18 from set 3: male TWAS). The female-specific modules and the male-specific modules, although containing different sex-specific TWAS genes (set2: female TWAS genes, and set 3: male TWAS genes), both converged on pathways associated with protein ubiquitination/mTOR/EIF2 signaling, estrogen receptor signaling, HIPPO signaling[56], and with placental growth factors associated with fibrosis[57] and epithelial-to-mesenchymal transition, a crucial process for trophoblast invasion[58]. Male-specific modules also showed an enrichment for inflammatory pathways, like IL15 production and signaling, while female-specific modules were also related to insulin signaling (Supplementary Data 28). Thus, the WGCNA analysis indicated that some placental schizophrenia risk genes may converge on biological processes and etiopathogenetic mechanisms, common to the two sexes (mTOR signaling), albeit not necessarily driven by the same genes; at the same time, other placental schizophrenia risk genes may converge on sex-specific processes (insulin signaling in females, and IL15 signaling in males) revealing a possible impact of the sex-biased architecture on placental gene coexpression (see Supplementary Data 28 for further details).

### A set of placental genes associated with schizophrenia is tissue-specific and not detected in midgestational prenatal cortex

The relationship between genetic variation and gene expression is often tissue-nonspecific.

Analyzing shared eQTL across tissues might facilitate the detection of genes associated with traits through multi-tissue TWAS approaches;[59] however, it can also impede the detection of genetic mechanisms that, by acting in specific organs and tissues, affect the risk of developing a particular disease. As a consequence, TWAS genetic associations that are not tissue-specific are less likely to identify candidate causal genes that also offer a suitable target for prevention and treatment. Because the aim of this research is to identify candidate causal genes of schizophrenia which, by acting in early life in placenta, may affect trajectories of risk for this disorder, we performed a TWAS analysis in fetal brain, and we prioritized the potential placental-specific schizophrenia TWAS genes, excluding from our list the TWAS-significant genes in prenatal cortex, as well as the genes with predicted expression associated with schizophrenia based on SMR analysis in adult and fetal brain and blood[10] (Supplementary Fig. 1).

Fastq files from 175 midgestational prenatal cortical samples[33] with good quality RNA (RIN > 6.9) were downloaded from the NCBI SRA public database (dbGAP access: phs001900) and re-aligned to the GRCh38. The transcriptomes of these prenatal brain samples were characterized by measurement of (i) gene-level expression and (ii) transcript-level quantification, as described for the placental transcriptome, resulting in 54,132 genes and 201,352 transcripts, which became respectively 18,580 and 73,085 after filtering lowly expressed features. Further quality check and outlier detection were performed as described for the placental transcriptome, leading to a final sample of 166 fetal brains (67 females) for the gene-level and the transcript-level data. Supplementary Data 29 summarizes sample demographics.

We detected 8879 genes and 32,887 transcripts (corresponding to 12,519 unique genes) with heritable cis-regulated expression in prenatal cerebral cortex. The TWAS highlighted 145 genes (Fig. 3a) and 411 transcripts (corresponding to 301 unique genes) (Fig. 3b) associated with schizophrenia (Bonferroni corrected $p$ value: $p_{corr} \leq 0.05$), so that there were 386 unique genes with predicted expression in prenatal cerebral cortex associated with schizophrenia at gene and/or transcript level (Supplementary Data 30–32). We also performed prenatal brain TWAS stratified by sex, detecting, respectively 355 and 345 unique genes with predicted expression in female (Supplementary Data 33–35) and male (Supplementary Data 36–38) prenatal cortex associated with schizophrenia at gene and/or transcript level, respectively. A comparison of the individual TWAS results in the two sexes indicated an overlap of 129/571 TWAS-significant ($p_{corr}<0.05$) genes (22.59%) in males and females (Supplementary Data 39), higher concordance compared with what we observed in placenta ($\chi^2 = 22.57$, $p < 0.0001$). When using a less stringent statistical threshold (FDR < 0.05) the overlap between TWAS-significant genes in males and females was slightly stronger, that is, 24.86 % (828/3331, Supplementary Data 40), and still higher compared with placenta ($\chi^2 = 25.45$, $p < 0.0001$); consistently, the correlation between the Z scores of the two analyses ($r = 0.239$ and $r = 0.161$ at gene and transcript level respectively) was higher in brain compared with placenta ($r = 0.1629$ and 0.0718).

By comparing the TWAS statistics for schizophrenia in placenta and in prenatal cortex, we found evidence of a very weak correlation, particularly at the transcript level ($r = 0.027$), and in the analyses stratified by sex ($r = 0.005$ in females and 0.017 in males; Fig. 3c–h). We leveraged the results of the TWAS in prenatal brain to prioritize the possible placenta-specific schizophrenia risk genes, that is, the genes whose significant association between predicted expression in placenta and schizophrenia (Bonferroni corrected $p$ value: $p_{corr} \leq 0.05$) could be considered specific, because they are not mirrored by an analogous relationship in prenatal cortex. To define placenta-specificity in a more conservative way, we used a less stringent statistical threshold (FDR < 0.01) in fetal brain to define TWAS-significance, thus decreasing the risk of considering, as placenta-specific, possible false negative TWAS genes in fetal brain. In this way, we prioritized 206 (150 SMR-validated) placental-specific genes associated with schizophrenia in placenta (whole sample, plus males or females) but not in our prenatal cortex samples (whole sample, plus males or females) (Fig. 3i) as candidate genes that may have a causative role on schizophrenia risk based on their role in placenta and not in prenatal cerebral cortex (Supplementary Data 41). We cross-checked our list of placental prioritized genes with a reference of prioritized schizophrenia genes from SMR analysis in adult and fetal brain and blood[10], and we detected only three common genes, which were excluded from our list of placental specific schizophrenia genes, leading to a priority set of 203 genes (147 SMR-validated, Supplementary Fig. 1).

Pathways analysis on the 203 placental-specific schizophrenia TWAS genes supported the enrichment for mTOR signaling, insulin, estrogen, and EIF2 signaling (Supplementary Data 41). We also

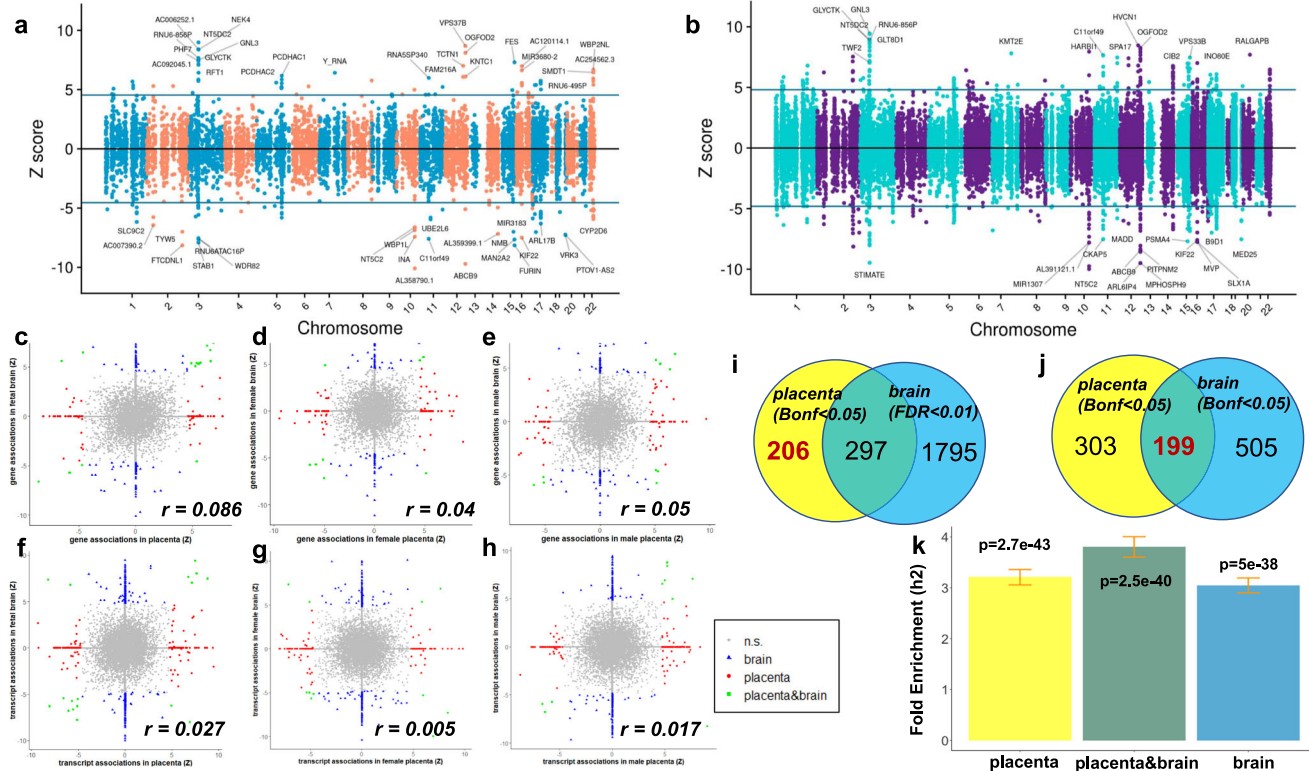

**Fig. 3 | Comparison between schizophrenia TWAS in placenta and in fetal brain.** TWAS identifies genes (**a**) and transcripts (**b**) whose cis-regulated expression in prenatal cortical brain is associated with schizophrenia. Scatterplots of the correlation between the Z scores from the TWAS analyses in placenta (x-axis) and fetal brain (y-axis) at gene (**c**–**e**) and transcript (**f**–**h**) level, in the whole sample (**c**, **f**), in female (**d**, **g**), and in male (**e**, **h**) placentae (N = 147: 73 females and 74 males) and prenatal cortical brain (N = 166: 67 females and 99 males). Genes and transcripts TWAS-significant only in placenta (red), only in fetal brain (blue), and in both organs (green) are highlighted. Venn diagrams of the overlap of schizophrenia TWAS significant genes in placenta (yellow circle) with genes associated with schizophrenia in fetal brain (azure circle) with FDR < 0.01 (**i**), and with Bonferroni-corrected p value < 0.05 (**j**): when using Bonferroni-corrected p value < 0.05 in both organs to define TWAS-significance, we detected 199 overlapping genes (placenta&brain pleiotropic), and 303 placenta-specific TWAS genes, that is, genes that

were TWAS-significant in placenta and not in fetal brain (**j**); the placenta-specific TWAS genes are less, i.e., 206, when using a more conservative criterion to define placenta-specificity, that is, using a more permissive threshold to define TWAS-significance in fetal brain (FDR < 0.01), thus reducing the risk of false negative in brain and false positive placenta-specific TWAS genes (**i**). **k** enrichment-fold (y-axis) for schizophrenia heritability among the SNPs associated with the expression of the TWAS genes associated with schizophrenia in placenta (left, yellow bar; N of genes = 502 from a TWAS analysis in 147 placentae), fetal brain (right, azure bar; N of genes = 704, from a TWAS analysis in 166 prenatal cortical brain), and in both placenta and fetal brain (center, green bar; N of genes = 199). Error bars represent standard error of enrichment fold, with mean as measure of center; p value of the enrichment fold computed by stratified LD score regression analysis is at the top of each bar.

detected 505 genes with TWAS association in prenatal cortex but not in placenta (Supplementary Data 42), and 199 genes associated with schizophrenia both in placenta and prenatal brain (Bonferroni corrected p-value: $p_{corr} \le 0.05$ in both organs) (Fig. 3j), suggesting that pleiotropic effects in the two organs may contribute to trajectories of risk for the disorder, also through tissue-specific mechanisms. Among the 199 placental genes also associated with schizophrenia in prenatal cortex, 142 show opposite sign of association in placenta and brain. Indeed, the placenta&brain schizophrenia TWAS genes imply an activation of insulin signaling in placenta and its inhibition in prenatal cortical brain, due to the presence of genes with opposite sign of association in the two organs (Supplementary Data 43).

### Genetic effects on the placental transcriptome contribute to the heritability of schizophrenia
Our TWAS results indicate that genetic risk for schizophrenia unfolds in part through placental gene expression and in part through fetal brain gene expression, with the two mechanisms not mutually exclusive. We, therefore, estimated whether genetic variants associated with the expression of schizophrenia risk genes in placenta (placental-eQTL) and fetal brain (fetal brain-eQTL) contribute disproportionally to the genetic underpinning

of the disorder. To this purpose, we performed stratified LD score regression (S-LDSC), a widely used tool that estimates the heritability ($h^2$) enrichment of a functional annotation, defined as the proportion of SNP-heritability explained by the annotation divided by the proportion of SNPs in the annotation. The proportion of $h^2$ explained by placental eQTL's was equal to 0.16 (standard error (s.e.) = 0.008), and the placental eQTL SNPs showed a 3.2 (s.e.= 0.15, p = 2.69e-43) enrichment-fold for SNP-heritability of schizophrenia. The fetal brain-eQTL SNPs explained a higher proportion of $h^2$ (0.20, s.e. = 0.010), as expected given the greater number of SNPs; however, the enrichment-fold for heritability of the fetal brain-eQTL SNPs was slightly lower compared with the placental, that is, 3.0 (s.e. = 0.14, p = 4.99e-38). Of note, eQTL SNPs associated with the expression of schizophrenia risk genes both in placenta and fetal brain showed the strongest fold-enrichment for heritability, that is, 3.8 (s.e.= 0.20, p = 2.50e-40) (Fig. 3k). Overall, these data indicate that a sizeable proportion of the heritability of schizophrenia is explained by genetic variation associated with the expression of risk genes in placenta, and support the often-overlooked possibility that genetic variation contribute to risk for schizophrenia also through pleiotropic effects in fetal tissues.

**Table 1 | Placental genes with TWAS-significant (Bonferroni-corrected p value < 0.05) association with schizophrenia and other disorders ant traits**

| | GWAS sample size (cases; controls) | GWAS loci | TWAS genes (g) | TWAS transcripts (tx) | TWAS unique genes (g&tx) | Reference (PMID) |
|---|---|---|---|---|---|---|
| ADHD | 55,374 (20,183; 35,191) | 12 | 2 | 10 | 10 | Demontis, Nat.Gen 2019 (30478444) |
| Autism | 46,350 (18,381; 27,969) | 5 | 2 | 0 | 2 | Grove, Nat.Gen. 2019 (30804558) |
| Schizophrenia | 130,644 (53,386; 77,258) | 222 | 100 | 235 | 262 | Trubetskoy, Nature 2022 (35396580) |
| Bipolar Disorder | 413466 (41,917; 371,549) | 64 | 22 | 72 | 75 | Mullins, Nat.Gen. 2021 (34002096) |
| Major Depression | 173,005 (59,851; 113,154) | 43 | 23 | 28 | 37 | Wray, Nat.Gen. 2018 (29700475) |
| Intelligence Quotient | 78,308 | 336 | 7 | 8 | 14 | Sniekers, Nat.Gen.2017 (28530673) |
| Insomnia | 113,006 | 3 | 0 | 0 | 0 | Hammerschlag, Nat.Gen. 2017 (28604731) |
| Birthweight | 321,223 | 145 | 90 | 180 | 204 | Warrington, Nat.Gen. 2019 (31043758) |
| BMI | ~700,000 | 940 | 22 | 35 | 52 | Yengo, HMG 2018 (30124842) |
| Height | ~700,000 | 3,289 | 93 | 169 | 197 | Yengo, HMG 2018 (30124842) |
| Type 2 Diabetes | 659,318 (62,892; 596,426) | 142 | 48 | 103 | 117 | Xue, Nat.Comm. 2018 (30054458) |

Column 2 shows the sample size of GWAS studies (disorders and traits in column 1) that provided the summary statistics for TWAS. Column 3 shows the number of GWAS loci significantly (p value < 5e-08) associated with each disorder or trait. Columns 4, 5 report the number of genes (column 4), transcripts (column 5) with TWAS-significant association with each disorder or trait. Column 6 reports the number of unique genes with TWAS-significant association with each disorder or trait, at gene and/or transcript level. Columns 7 report the article providing the reference of the GWAS summary statistics for the TWAS.

## The placental transcriptome is highly and selectively relevant for schizophrenia, in comparison with other developmental disorders and traits

Part of the genomic component of schizophrenia risk is shared with other complex disorders and traits, some of which also have developmental origins. We asked whether placental genes associated with schizophrenia would show relationships with other disorders and traits. We, therefore, performed analogous placental g-TWASs and tx-TWAS on other complex disorders and traits with a developmental component. These included autism, ADHD, bipolar disorder, major depression, intelligence quotient (IQ), fetal birthweight, body mass index (BMI), height, and type 2 diabetes (T2D); we also performed placenta TWAS on insomnia, a condition with a less apparent developmental component.

We detected genes and transcripts with a TWAS-significant association in placenta with each of these conditions, with the exception of insomnia (Table 1, Supplementary Data 44–53), though the results underscore the priority of placenta gene expression for risk for schizophrenia. As expected, the number of TWAS-significant genes was moderately correlated with the number of GWAS-significant loci ($r = 0.39$), which was in turn correlated with the sample size of the GWAS ($r = 0.61$). Of note, schizophrenia is the disorder with the greatest number of placental TWAS-significant genes and transcripts, even though other disorders and traits (IQ, BMI, height) have higher numbers of GWAS-significant loci (Table 1). Indeed, schizophrenia has the greatest number of placental TWAS-significant genes ($t = 2.67$, $p = 0.0322$) and transcripts ($t = 3.11$, $p = 0.0171$) compared with the others, after adjusting for the number of GWAS hits and GWAS sample size. Further, the ratio between placental TWAS-significant hits and GWAS-significant loci is relatively higher for schizophrenia, major depression, bipolar disorder, birthweight, and type 2 diabetes (range 0.63–1.08) than for IQ, BMI, height (range 0.03–0.05). ADHD and autism are respectively associated with nine and two placental unique genes; however, the smaller sample size of these respective GWASs does not allow exclusion of power issues in affecting the results (Table 1, Supplementary Data 44–53).

We explored the overlap between the placental genes associated with schizophrenia and the other disorders and traits with Bonferroni-significant p value, and we also used a threshold-free algorithm to detect concordant and discordant patterns of TWAS-associations with

potential biological relevance[60]. The placental TWAS associations show a pattern of relatively limited concordance between schizophrenia and autism ($r = 0.1079$, Fig. 4a), and the two placental genes associated with autism were also associated with schizophrenia with nominal levels of significance (RPS15AP1: $p = 0.001$, FDR = 0.028; RP1-198K11.5: $p = 0.046$, FDR = 0.061, Supplementary Data 44−54). As expected, there was also limited concordance between schizophrenia and ADHD ($r = 0.0705$), particularly for downregulated genes (Fig. 4b), and six (MAD1L1, KDM4A, IPO13, ATP6V0B, DPH2, PTPRF) of the nine placental genes associated with ADHD were also significantly associated with schizophrenia (Supplementary Data 45, 54). Among these genes, MAD1L1 was also positively associated with depression. The placental transcriptomic associations with schizophrenia showed some degree of concordance with depression ($r = 0.148$, Fig. 4c), and 11 genes were significantly associated with both schizophrenia and major depression. Of note, however, 5 of these 11 placental genes (CLP1, MARK3, KLC1, TRMT61A, APOPT1) were associated with opposite sign between the two disorders, suggesting that both an up-regulation and a down-regulation in placenta of these genes may bias towards neurodevelopmental trajectories of risk for psychiatric disorders (Supplementary Data 46, 54). As expected, the placental transcriptomic associations with schizophrenia and bipolar disorder did show positive correlation of TWAS Z-scores ($r = 0.451$, Fig. 4d), and 30 genes were significantly associated with both schizophrenia and bipolar disorder. 4/30 placental genes (TSTA3, BRD8, WDR82, SPG7) that were associated with both schizophrenia and bipolar disorder had opposite signs (Supplementary Data 47, 54), so they were predicted to be up-regulated in the placentae of individuals who developed one disorder and down-regulated in the other disorder.

Interestingly, the TWAS Z-scores of schizophrenia and birthweight were not correlated ($r = −0.01$) but, when considering the absolute values of the Z-scores, we did detect a positive correlation ($r = 0.10$). Indeed, the overlap between schizophrenia and birthweight involved placental genes with changes in the same (nine genes at $p_{corr} < 0.05$) and opposite (six genes) direction associated with the two conditions (Fig. 4e, Supplementary Data 48, 54), consistent with data showing that both high and low birthweight represent a risk factor for schizophrenia[3,61,62]. Overall, these data suggest that an effect of placental genes on fetal growth may lead to very different neurodevelopmental trajectories, likely depending on the various factors that can

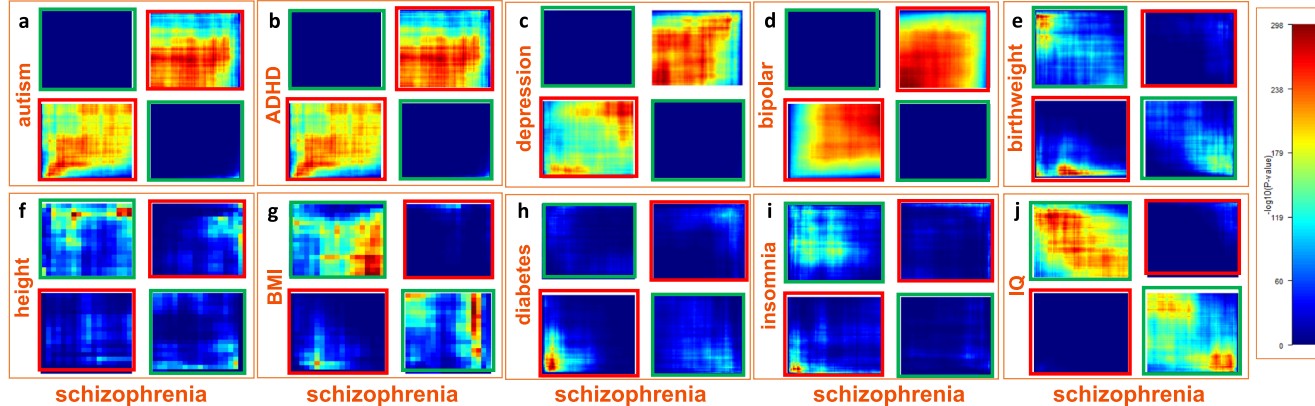

**Fig. 4 | Concordant and discordant placental gene expression signatures between schizophrenia and other disorders and traits.** RRHO2[61] heatmaps showing concordant (bottom-left and top-right quadrant in each panel, highlighted in a red frame) and discordant (top-left and bottom-right quadrant in each panel, highlighted in a green frame) placental TWAS association ($N = 147$) with schizophrenia with placental TWAS association with autism (**a**), ADHD (**b**), major depression (**c**), bipolar disorder (**d**), birthweight (**e**), height (**f**), body mass index (BMI) (**g**), type-2 diabetes (**h**), insomnia (**i**), and Intelligence Quotient (IQ) (**j**). Concordant and discordant associations are estimated using a threshold-free algorithm based on a rank-rank hypergeometric overlap approach[61]. The heatmaps for the comparison of each disorder and trait with schizophrenia are separated into rectangles with orange lines. Color bar represents negative logarithm of $p$ value of the overlap from the hypergeometric test.

be associated with birthweight, that is, birth asphyxia[63], gestational diabetes, and other early life complications.

A link between placental schizophrenia risk genes and impaired growth-related processes was also suggested by the discordance of the TWAS associations with schizophrenia with height ($r = -0.06313$, Fig. 4f, Supplementary Data 49, 54) and BMI ($r = -0.07834$, Fig. 4g, Supplementary Data 50, 54). We also detected concordant (three genes at $p_{corr} < 0.05$) and discordant (six genes at $p_{corr} < 0.05$) patterns of placental TWAS associations between schizophrenia and type 2 diabetes ($r = -0.003$; with absolute Z-scores $r = 0.086$, Fig. 4h, Supplementary Data 51, 54), as well as between schizophrenia and insomnia when considering associations that were not TWAS-significant ($r = -0.004$; with absolute Z-scores $r = 0.05$, Fig. 4i, Supplementary Data 52). As expected, the placental TWAS association with schizophrenia and IQ was discordant ($r = -0.0929$, Fig. 4j, Supplementary Data 53, 54) and three genes (*SNU13, WBP2NL, RP1-257I20.14*) were significantly associated with schizophrenia and IQ with opposite sign.

Finally, these analyses indicated overall that a large proportion of placental TWAS-significant genes were specifically associated with schizophrenia alone. In particular, 188 out of the 203 genes associated with schizophrenia in placenta, and not in fetal brain, did not show any significant relevant association with any of the other disorders and traits analyzed. We prioritized these 188 genes as placenta-schizophrenia-specific risk genes, 139 of which were SMR-validated (Supplementary Data 54, Supplementary Fig. 1).

### Placental risk genes with predicted higher expression in schizophrenia are enriched in extravillous trophoblast

The placenta is composed of different compartments and heterogeneous cell types which contribute in different ways to its development and functionality. Because our TWAS study leverages transcriptome data obtained from bulk placental tissue, we used publicly available single-cell data to explore whether the placental TWAS associations point to a specific placenta cell type. We analyzed single-cell data from placenta obtained at the end of the 1st trimester[64], when all the cell types of the term placenta are already present (Fig. 5a). Using the Bioturing BBrowser (version 2.10.40), we generated two signature scores summarizing the expression of the placental schizophrenia risk genes with predicted higher and lower expression associated with schizophrenia. Placental genes predicted to be

downregulated in schizophrenia were particularly enriched in syncytiotrophoblasts (STB, $t = 64.08$, $p < 2.2e{-}16$) (Fig. 5b), while placental genes predicted to be upregulated in schizophrenia were enriched in villous trophoblast (VCT, $t = 20.93$, $p < 2.2e{-}16$), extravillous trophoblast (EVT, $t = 6.80$, $p = 1.07e{-}11$), and in Hofbauer cells (HBCs, $t = 30.61$, $p < 2.2e{-}16$), fetal-origin macrophages residing in the placenta that are likely involved in responding to placental infection and protection of the developing fetus[65] (Fig. 5c). Because of previous evidence indicating a relative up-regulation of genes in schizophrenia risk loci in placentae from male compared with female offspring[35], we analyzed whether the expression of the placental specific schizophrenia risk genes was biased in specific cell types based on sex. Interestingly, while we did not detect significant differences in the expression of the genes predicted to be down-regulated in schizophrenia (Fig. 5d), we found that the genes with predicted higher placental expression in schizophrenia were relatively up-regulated in EVT ($t = 7.11$, $p = 2.3e{-}12$) in males compared with females, while this difference was less evident in VCT ($t = 1.97$, $p = 0.049$), and STB ($t = 1.51$, $p = 0.13$) (Fig. 5e).

### Placental schizophrenia risk genes are upregulated in placentae from pregnancies with SARS-CoV-2 maternal infection

Our pathway analysis indicated a link between the predicted gene expression changes associated with schizophrenia in placenta and the activation of the Coronavirus pathogenesis pathway. The placental schizophrenia risk genes enriched in this pathway are *ATF4, FURIN, IRF3, MAPK3, RPS10,* and *RPS17*. We, therefore, explored the possibility of a convergence between genomic risk for schizophrenia and maternal SARS-CoV-2 infection during pregnancy in affecting gene expression changes in placenta. In particular, we analyzed whether placental schizophrenia risk genes were enriched among the differentially expressed genes in term placental villi from pregnancies complicated with SARS-CoV-2 infection ($N = 5$) compared with placental villi from uninfected control individuals matched for maternal age, gestational age, maternal comorbidities, and mode of delivery ($N = 3$)[66]. We found that the placental schizophrenia risk genes are differentially expressed, and in particular up-regulated ($p = 1.06e{-}19$ for the full set of the placental TWAS schizophrenia genes, Fig. 6a; $p = 6.68e{-}04$ for the 139 prioritized genes), in placentae from SARS-CoV-2 positive pregnancies, supporting the possibility that the immune reaction associated with this maternal infection may induce transcriptomic changes in placenta potentially relevant for

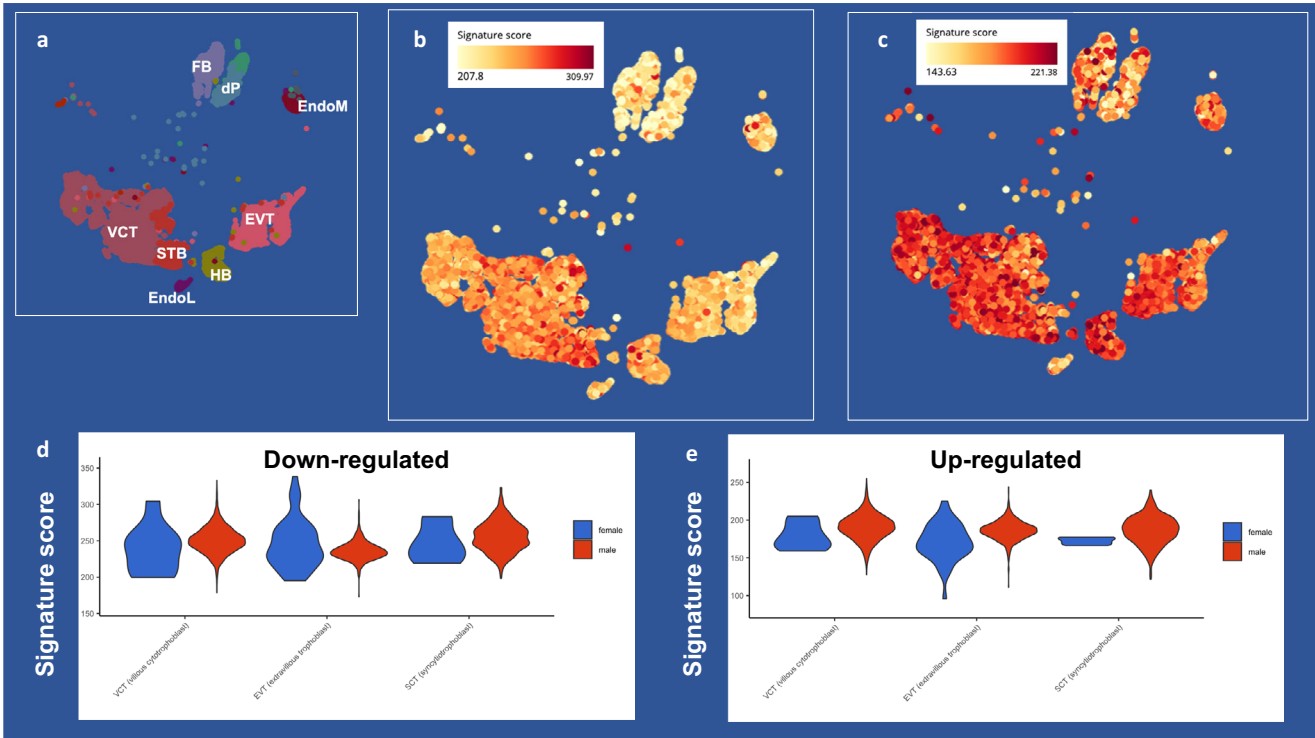

**Fig. 5 | Single-cell enrichment of placental schizophrenia risk genes.**
**a** Scatterplot of placental fetal cells detected at the maternal-fetal interface in 1st-trimester placenta data[65] (provided for comparison with **b** and **c**) [VCT = villous trophoblast, EVT = extravillous trophoblast, STB = syncytiotrophoblast, HBC = Hofbauer cells, EndoM and EndoL = endothelial cells, dP = perivascular cells, FB = fibroblasts]. scatterplots of the enrichment in placental cell types of the signature score summarizing the expression of the placental TWAS genes ($N = 502$) whose lower expression (**b**) and higher expression (**c**) is predicted to be causative of schizophrenia. Placental genes predicted to be down-regulated in schizophrenia are expressed in STB, while placental genes predicted to be up-regulated in schizophrenia are also expressed in VCT, EVT, and HBCs. Color bar represents expression enrichment (yellow: lower; orange/red: higher expression). violin plots of the signature scores (y-axis) summarizing the expression of the placental TWAS genes, whose lower expression (**d**) and higher expression (**e**) is predicted to be causative of schizophrenia, in VCT, EVT, and STB from female (blue) and male (red) placentae. See main text for statistics.

schizophrenia risk. Among the other psychiatric disorders, placental risk genes for bipolar disorder were also enriched among the genes up-regulated in placentae from SARS-CoV-2 pregnancies, but less significantly compared with schizophrenia, as were placental risk genes for BMI, birthweight, height, and type 2 diabetes (Fig. 6b). No significant enrichment was detected for placental risk genes for ADHD, ASD, IQ, insomnia, and depression. The enrichment of schizophrenia risk genes was also stronger compared with a set of immune genes, 11 of which were also in the list of TWAS hits for schizophrenia (Fig. 6b,c). Of note, in this dataset the schizophrenia risk genes tended to be highly ranked among the genes highly coexpressed with specific immune relevant genes, that is, heat shock proteins ($p = 3.87\text{e-}15$), NFKB ($p = 1.23\text{e-}06$), nuclear receptors ($p = 2.83\text{e-}07$), and blood-brain barrier genes ($p = 0.0002$) (Fig. 6d,e).

**Colocalization and replication analyses support the relationship between the placental transcriptome and genomic risk for schizophrenia**

As a further approach to prioritize placental genes with a possible causative role for schizophrenia, and to identify potentially causative variants acting in placenta, we performed a colocalization analysis[67,68], in the whole sample and in placentae from female and male offspring. This approach aims at identifying overlapping causal genetic variants for both molecular and complex traits. In addition to variant-level colocalization, we performed locus-level colocalization analysis. The latter approach has been recently shown to improve not only specificity but also sensitivity when using both TWAS and colocalization to detect biologically relevant genes with a possible causative role on disease[68]. We identified 48 placental genes with gene variant-level

colocalization probability (GRCP) and/or gene locus-level colocalization probability (GLCP) $\geq 0.50$, and 113 genes with GRCP/GLCP $\geq 0.10$, a higher number compared with the schizophrenia risk genes prioritized by colocalization analyses in larger datasets from different tissues[29,69]. Of note, 33 of these genes overlap the 139 placenta- schizophrenia-specific risk genes, which show TWAS-significant association with schizophrenia in placenta, and not in fetal brain, are SMR-validated and did not show any significant relevant association with any of the other disorders and traits analyzed. We define these 33 genes as the top-prioritized placenta- schizophrenia-specific risk genes (Supplementary Data 54). 13 and 14 of these genes share, as master regulators[70], WNT7 and IL-2R respectively (Supplementary Data 55a), which are known activators of the mTOR pathway[71,72]. More in general, many of these genes code for proteins with a role linked with the nutrients sensing capabilities of placenta, cell growth and response to stress (see also Supplementary Note 1 and Supplementary Data 55a,b-61).

These results add support to the conclusion that the placental transcriptome mediates at least in part the genomic risk for schizophrenia. To go a step further, we also performed a replication TWAS, leveraging transcriptomic and genomic data from 70 placentae (see Supplementary Data 62 for sample info) from the Eunice Kennedy Shriver National Institute of Child Health and Human Development (NICHD) Fetal Growth Study[73] (dbGaP accession: phs001717.v1.p1). Despite the smaller sample size compared with our discovery sample, and some differences in the RNA processing pipeline, we found significant positive and concordant associations between the Z scores from the two TWAS analyses in the discovery and in the replication datasets, in the whole replication sample ($N = 70$, $t = 23.40$, $r = 0.36$, $p < 2\text{e-}16$; Supplementary Fig. 3a,b), in female placentae ($N = 35$,

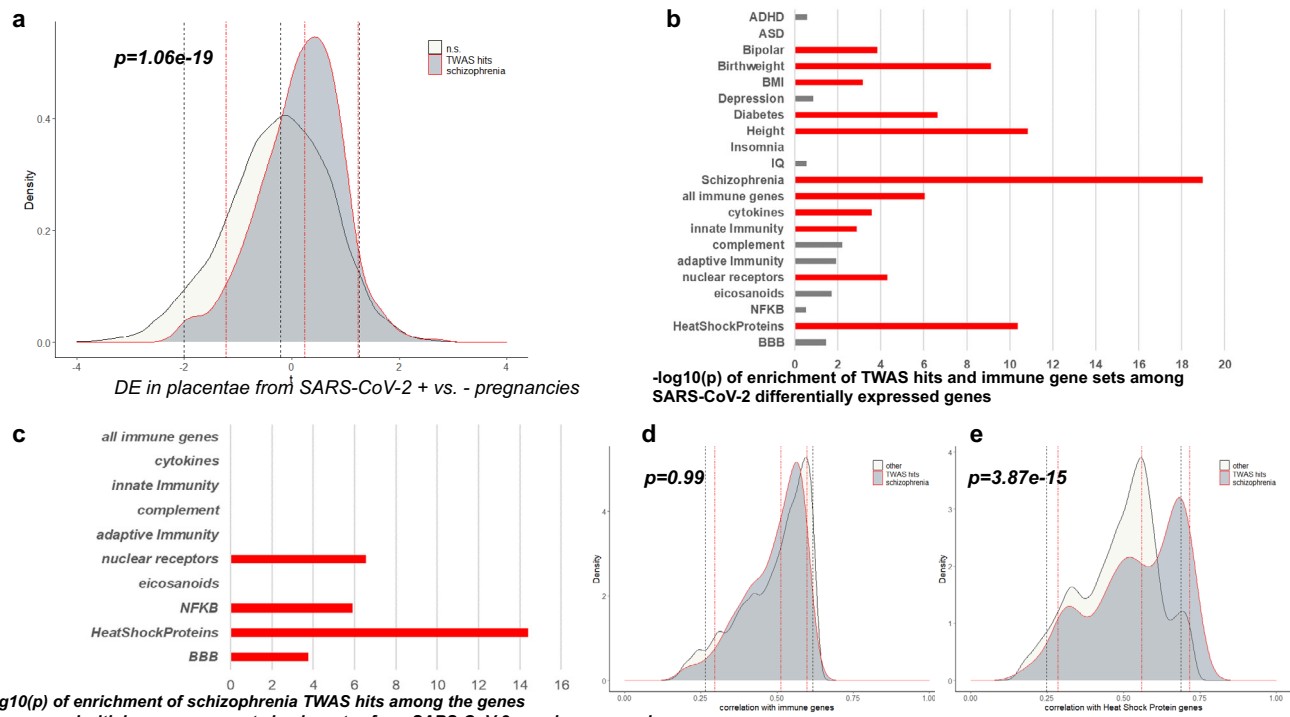

**Fig. 6 | Placental schizophrenia-risk genes and SARS-CoV-2 infection. a** Density plots of the t-statistics, from the differential expression analysis comparing placentae from SARS-CoV-2 positive ($N = 5$) and negative ($N = 3$) pregnancies, of placental schizophrenia-TWAS genes (TWAS hits, dark gray area under solid red curve), and all other genes (under black curve) (negative t-statistics = less expressed in placentae from SARS-CoV-2 positive pregnancies; and positive t-statistics = more expressed in placentae from SARS-CoV-2 positive pregnancies). Bars depict negative logarithm of the $P$ values of the enrichment, from the Wilcoxon *geneSetTest*, of TWAS hits for developmental disorders and traits, and immune gene sets, among the SARS-CoV-2 differentially expressed genes (**b**), and of the schizophrenia TWAS hits among the genes coexpressed with immune gene sets (**c**); orange bars indicate gene sets significantly enriched ($p < 0.05$) after Bonferroni-correction for multiple comparisons. Density plot of the absolute values of correlation coefficients of placental schizophrenia-risk genes (dark gray area), and all other genes with the immunity genes (**d**) and the heat-shock protein genes (**e**). In (**a**, **d**, and **e**) dotted lines depict 95% confidence intervals and median of the moderated t-statistics from the differential expression analysis (**a**) or the correlation coefficients (**d,e**) of the placental schizophrenia risk genes (red double-dotted lines) and of all the other genes (black dotted lines); one-sided $P$ value at the top of the graphics are from the Wilcoxon *geneSetTest*.

$t = 22.24$, $r = 0.34$, $p < 2e$-16; Supplementary Fig. 3c,d), and in the male sample ($N = 35$, $t = 24.24$, $r = 0.37$, $p < 2e$-16; Supplementary Fig. 3e,f). The correlation between Z-scores was stronger when considering only TWAS-significant genes in the discovery sample, particularly in placentae from male offspring ($r = 0.51$, 0.40, and 0.73, in the whole sample, in females, and in males, respectively). Consistently, the level of replication ($p < 0.05$) of the TWAS significant associations in our discovery sample was higher than expected by chance in the whole sample ($\chi^2 = 29.79$, $p < 0.0001$), and in placentae from female ($\chi^2 = 7.82$, $p = 0.005$), and from male ($\chi^2 = 61.02$, $p < 0.0001$) offspring (see Supplementary Note 2 and Supplementary Data 63–65 for detailed results).

## Discussion

Leveraging the statistical techniques of TWAS, SMR, and colocalization, in term placenta tissue, we integrated transcriptomic and genetic data with GWAS to detect and prioritize candidate causal genes for schizophrenia that, by playing a specific role in placenta, may contribute to the etiopathogenesis of schizophrenia. Most of these genes did not show analogous effects on risk of other developmental disorders and traits and were not identified as potentially causative based on similar analyses in mid-gestational fetal brain, suggesting a specific etiological role for the placenta in schizophrenia risk. TWASs in the whole sample and stratifying by sex ultimately highlight 188 placenta-specific schizophrenia risk genes, 139 of which are SMR-validated. Analysis at the transcript level also detects gene isoforms not associated with schizophrenia at the gene level, thus implicating splicing

events in the placental transcriptome that do not seem biased by sex-related differences, as has been reported in fetal brain[33]. Remarkably, the enrichment fold for heritability of schizophrenia explained by placenta eQTLs was slightly greater than that explained by fetal brain eQTLs.

The candidate molecular mechanisms through which genetic risk may impact placenta biology converge in indicating a link between schizophrenia genetic risk and the nutrient-sensing capabilities of placenta. In particular, the candidate causal genes linked with schizophrenia risk in placenta, both in males and in females, are associated with an inhibition of mTOR signaling, which represents a central hub in responding to multiple growth-related signals, including amino acids, glucose, oxygen, folate, and growth factors, to regulate trophoblast mitochondrial respiration, nutrient transport, and protein synthesis, thereby influencing fetal growth[74]. These results are supported by the colocalization analyses highlighting 33 top-prioritized genes, many of them implicated in biological functions linked with nutrient sensing and growth. Consistently across sexes, we also detect relative inhibition of EIF2 signaling, implicated in regulating protein translation in the endoplasmic reticulum stress response[75]. Thus, a scenario emerges in which genetic risk for schizophrenia may impact the ability of the placenta to integrate maternal and fetal nutritional cues with information from intrinsic nutrient-sensing signaling pathways, with effects on fetal growth and development[76]. This possibility is consistent with our previous findings of an interaction between genomic risk for schizophrenia and ELCs affecting trajectories of altered postnatal neurodevelopment[35,36]. Nutrient sensing and transport in the placenta

is strictly linked with trophoblast invasiveness[1] and the fetal stress response[77]. Indeed, schizophrenia risk is associated with lower expression of genes enriched in STB, the cells which form the main site of nutrient exchange at the maternal-fetal interface[78], and with higher expression of genes enriched in EVT, the cellular type that supports pregnancy by invading the maternal uterus, and particularly in males; consistently, defective invasion by EVT is the basis of many ELCs that are associated with schizophrenia[1].

One of the surprising results of the biological interpretation of the candidate placenta risk genes was activation of coronavirus pathogenesis. This led us to investigate expression of these genes in placenta from SARS-CoV-2-positive pregnancies. This analysis revealed upregulation of the set of schizophrenia-candidate causal genes in the coronavirus positive cases, suggesting that SARS-CoV-2 infection during pregnancy may be an environmental risk factor for schizophrenia. This is consistent with evidence of increased risk associated with other viral infections[3] and by recent findings on neurodevelopmental outcomes and SARS-CoV-2 infection during pregnancy[79]. More generally, the up-regulation of this gene set in placentae exposed to maternal SARS-CoV-2 infection, arguably a potentially serious complication, further supports the possibility of a convergence between ELCs and genomic risk for schizophrenia.

Of note, the TWAS by sex reveals a greater discordance of schizophrenia risk genes between the sexes in placenta as compared with fetal brain. This evidence argues that the genetic seeds of the final sex-biased outcome in the incidence of schizophrenia may be prominently planted in placenta. Apart from being enriched in mTOR and EIF2 signaling in both sexes, the schizophrenia risk genes are also associated with an alteration of insulin signaling in females and with inflammatory signaling in males. The involvement of either pathway implies that genetic risk affects the nutrient-sensing capabilities of placenta; however, this would happen in different molecular contexts, so that the involvement of one (insulin signaling in females) or the other (inflammatory signaling in males) may respectively compensate or exacerbate the impact on placental growth. This is consistent with the higher expression of schizophrenia risk genes in male EVTs, whose invading capabilities are sensitive to inflammation and oxidative stress[80], and with multiple evidence showing that males are more vulnerable than females to in-utero insults[81], also related to nutrient sensing and glucose metabolism alterations[81,82], and that genetic risk factors[35,36] might lead to alteration in brain development via placenta.

Notably, the absence of TWAS significance in mid-gestational fetal brain tissue among placenta-specific schizophrenia risk genes does not exclude that they also act in fetal or adult brain, perhaps in other brain regions, in specific cell types, or other developmental periods. However, the consistent evidence in placenta and the relative lack thereof in fetal brain make a strong case for a role mediated in large part by their biological effects in placenta. This is further supported by the fact that the placenta is the only human organ that is almost wholly uninnervated[83], largely excluding the possibility that the data are confounded by pleiotropy in nervous tissues. We leveraged the analysis in prenatal cortical brain not only to prioritize the placenta-specific schizophrenia risk genes, but also to highlight 199 genes whose predicted expression in both placenta and brain may contribute to shape trajectories of risk for schizophrenia. Interestingly, the expression changes of some of these genes are opposite in placenta and brain, e.g., associated with the activation of insulin signaling in placenta and its inhibition in prenatal cortical brain, suggesting that genetic risk for schizophrenia may compromise the ability of placenta to provide the first line of defense against metabolic stress by decreasing its own demands for glucose[84]. The strongest fold-enrichment for heritability of the eQTL SNPs associated with the expression of schizophrenia risk genes both in placenta and fetal brain support the often-overlooked possibility that genetic variation contributes to risk for schizophrenia also through pleiotropic effects in different organs.

A largely unanswered question is the degree to which the placental transcriptome is specifically linked with schizophrenia or whether its alterations may affect neurodevelopment in a more general way, and vulnerability to other psychiatric disorders. Although we detect placental genes relevant for other disorders and traits, and particularly for depression, bipolar disorder, birthweight, and diabetes, we also found that the association of the placental transcriptome is much greater for schizophrenia, compared with the other analyzed disorders and traits. Indeed, schizophrenia has the highest number of placental risk genes considering the number of GWAS hits and GWAS sample size. In particular, we do not find similar strong evidence for potential causative genes in placenta as risk factors for other neurodevelopmental disorders that have been associated with complicated pregnancies, including autism and ADHD, but we cannot exclude the possibility that investigating gene expression in placentae of earlier gestational age may detect stronger signals for other neurodevelopmental disorders.

A limitation of our study is that the identification of placental candidate risk genes relies only on statistical associations between predicted gene expression and schizophrenia. Such limitation is inherent in all studies based on the integration of GWAS with expression mapping[21–26], which have identified candidate causal genes for many disorders. Our study focuses on identifying candidate causal genes that act in placenta, a crucial organ in regulating early brain development. To date, only one TWAS has been previously performed in placenta, on tissue from extremely preterm babies, using a RNAseq library preparation that allows the generation of only one 50 bp fragment for transcript[85]. As expected because of low sensitivity, this study identified only seven placental genes associated with schizophrenia, three of which are nonetheless replicated in our dataset (COQ10B, placenta-specific, and VPS29 and SEC11A, also associated with schizophrenia in brain).

Our study highlights the relevance of placenta biology in affecting early trajectories of risk for schizophrenia, moving a step forward in the search and prioritization of placental genes with a potential role in schizophrenia pathogenesis, and in the understanding of the link between placenta, early brain development and schizophrenia. Moreover, this line of research may help to identify strategies of postnatal prevention benefiting subjects at higher risk, given that alterations of the placental risk genes can be detected decades before the possible onset of the disorder, i.e., at birth, when the placenta is available[86], or even during pregnancy, detecting placenta-derived molecules in maternal blood;[87] in this regard, the placental TWAS schizophrenia genes ZNF664 (prioritized), KLC1, MAN2A2, CTDSPL, and MSI2 have been detected in maternal blood as cell-free RNAs potential useful for the prediction of pregnancy outcomes[87]. Finally, the study of placental mechanisms with a potential role in schizophrenia pathogenesis may help to identify subsets of patients who may have a different clinical course and response to treatment, which could be tailored to their etiopathogenetic mechanism.

In conclusion, our findings, while not detracting from the importance of gene expression in brain for schizophrenia risk, reveal a larger picture that includes placenta: both placenta and brain might contribute to early and reversible trajectories of risk for the disorder, but most research on brain development has been exclusively focused on the brain. Neglecting the investigation of placental mechanisms of risk may miss relevant opportunities for prevention.

## Methods
### Sample and RNA sequencing data processing
Placenta tissues were collected as part of the Rhode Island Child Health Study (RICHS)[88]. This population consists of singleton, term infants (gestational age: 37 – 41 weeks) born without serious pregnancies complications or congenital or chromosomal abnormalities. RNA sequencing and genotyping was performed as described

elsewhere[42,89]. Fastq files from 150 placental samples previously selected as having good quality RNA sequencing and genotyping data[42] were downloaded from the NCBI SRA public database (SRP095910; dbGaP access: phs001586.v1.p1) and were aligned to the Genome Reference Consortium Human Build 38 (GRCh38). The transcriptomes of these placentae were characterized by measurement of features expression: (i) gene-level expression as Reads Per Kilobase Million (RPKM) and (ii) transcript-level quantification, which relies on existing gene transcripts annotation, as Transcript Per Million (TPM), using the SPEAQeasy pipeline[90], resulting in 58,037 genes and 198,093 transcripts. After further quality check we dropped one sample from our analyses, because it had low overall alignment rate with high chrM alignment rate. To detect genes and transcripts reasonably expressed in placental tissue, we filtered lowly expressed features using the *expression_cutoff* function in the jaffelab R package[91] to establish the lower values of median expression allowed. Thus, we retained 21,115 genes (median RPKM ≥ 0.285) and 53,637 transcripts (median TPM ≥ 0.21) corresponding to 17,874 unique genes. Expression values were then $\log_2$ tranformed with an offset of 1 and Inter Array Correlation was used to identify subject outliers[92,93]. Briefly, we computed samples' pairwise correlations (Pearson's $r$) to define relative Euclidean distance among subjects. Then, we excluded outliers when the deviation from the average distance is ≥3 standard deviations, resulting in 146 samples (73 females) for the gene level and 147 samples (73 females) for the transcript-level data. Supplementary Data 1 summarizes sample demographics. Finally, we adjusted features expression using the *empiricalBayesLM* function in the WGCNA R package[94] accounting for ten genomic principal components, mitochondrial RNA rate, ribosomal RNA rate, gene assignment rate, and RNA sequencing batch, retaining the effect of sex. Adjusted (residual) gene and transcript expression was used for subsequent analyses.

The same pipeline was used to obtain RNA sequencing data from prenatal cortical brain. Fastq files from 175 midgestational prenatal cortical brain samples[33] with good quality RNA (RIN > 6.9) were downloaded from the NCBI SRA public database (dbGAP access: phs001900.v1.p1) and re-aligned to the GRCh38. The transcriptomes of these fetal brain samples were characterized by measurement of (i) gene-level expression and (ii) transcript-level quantification, using the SPEAQeasy pipeline[90], resulting in 54,132 genes and 201,352 transcripts, which became respectively 18,580 and 73,085 after filtering lowly expressed features. Further quality check and outlier detection were performed as described for the placental transcriptome, leading to a final sample of 166 fetal brains (67 females) for the gene-level and the transcript-level data. Supplementary Data 29 summarizes sample demographics.

The TWAS replication analysis was performed using genotype and RNA sequencing Fragments Per Kilobase Million (FKPM) data downloaded from the NCBI dbGaP database (Study Accession: phs001717.v1.p1) from 74 placental samples, from the Eunice Kennedy Shriver National Institute of Child Health and Human Development (NICHD) Fetal Growth Study[73]. RNA sequencing data were available only at the gene level. Quality check and outlier detection were performed as described for the placental (discovery) and fetal brain transcriptomes, leading to a final sample of 70 placentae (35 females). Supplementary Data 62 summarizes replication sample demographics.

## Genotyping
Genomic DNA was extracted and genotyping and imputation were performed as described elsewhere for placenta[42,89], fetal brain[33], and for the replication dataset[73]. To detect population stratification structure, we performed strict QC on genotype data. SNPs with genotype missing rate >1%, deviation from Hardy-Weinberg equilibrium ($p < 0.001$), or minor allele frequency <5% were removed. Relatedness was investigated with identity by descent (IBD) test using the PLINK[95]

genome option. Independent single-nucleotide polymorphisms (SNPs) were obtained using LD pruning[96] with 200 KB windows and 100 variant step sizes. One of each pair with $R^2 > 0.2$ was removed until there were no paired SNPs with $R^2 > 0.2$. Independent SNPs were used to perform principle components analysis with *smartpca* in EIGENSOFT[97] (to obtain multidimensional scaling components for quantitative measures of ancestry). In eQTL analysis, we removed SNPs that had a genotype missing rate >10%, deviation from Hardy–Weinberg equilibrium ($p < 1e06$), or minor allele frequency <0.05%. All the individuals included in the analysis had overall genotyping rate higher than 97%.

## Transcriptome-wide association study
TWAS was performed using both adjusted gene-expression (g-TWAS) and transcript-expression (tx-TWAS) as reference panels, and GWAS summary statistics of schizophrenia from the latest release from the Psychiatric Genomic Consortium[10]. Because our placental sample was composed mainly of individuals who self-identified as white, we used the GWAS summary statistics from the PGC3 European cohort. We performed fusion TWAS, as described at http://gusevlab.org/projects/fusion[22]. LD was considered using the European ancestry samples of the 1000 Genome phase III aligned to the GRCh38 as reference panel. For each gene, the relationship between Single Nucleotide Polymorphism (SNP) genotypes in the 500-kbp locus and gene expression was modeled in five-fold cross-validation using multiple approaches: lasso, elastic net and top-SNP (expression Quantitative Trait Loci, eQTL). The tagging SNP (eQTL identifier), the best-performing model, the effect size ($R^2$) and the model p-value are reported. The SNP-expression weights from the best-performing model were retained to estimate the genetic effect on gene expression (eQTL Z-value), which was combined with the genetic risk (eQTL GWAS-derived Z-value) to obtain the TWAS statistic (Z-value)[22,98]. Positive Z-values are suggestive of a link between over-expression and disease risk and negative Z values between under-expression and disease risk. We used GCTA[99] to estimate cis SNP heritability for each gene in our dataset, and analysis was restricted to those exhibiting significant heritability (cis-$h^2$ $p < 0.1$, in light of the sample size). Meaningful eQTL were calculated for 8558 genes and 18,582 transcripts (8523 unique genes), which provided the multiple comparisons for Bonferroni correction (respectively, g-TWAS: $p_{corr} = 0.05/8558 = 5.84 \times 10^{-6}$ and tx-TWAS: $p_{corr} = 0.05/18,585 = 2.69 \times 10^{-6}$). Because in the MHC genomic region, the high LD may enhance false discovery[43], we report separately the TWAS results for MHC region (131 genes and 251 transcripts with 136 unique genes) allowing the same statistical thresholds (Supplementary Data 66–71). To investigate the specificity of the relationship with schizophrenia of placenta genes and transcripts, we also computed TWAS using GWAS statistics from other major psychiatric disorders (Bipolar Disorder [BP][100], Major Depression Disorder [MDD][101], Autism Spectrum Disorder [ASD][102] and Attention Deficit Hyperactivity Disorder [ADHD][103], psychiatric-related phenotypes (Intelligence Quotient [IQ][104] and Insomnia[105]) and other traits and diseases whose risk is known to be associated with pregnancy complications (Fetal Birth Weight [Fetal BW][106]), Body Mass Index [BMI][107], Height[107] and Type II Diabetes [T2D][108]. We explored the correlation (Pearson's $r$) of TWAS Z values among diseases and investigated the overlap of significant genes, which may underlie shared pathways of risk. We used a threshold-free algorithm, RRHO2, based on a rank-rank hypergeometric overlap approach[60], to detect pattern of concordant and discordant TWAS association of schizophrenia with the other disorders and traits, and to compare TWAS results in the discovery and replication sample.

## Summary-based Mendelian Randomization (SMR)
We performed placental SMR analysis[25], for schizophrenia, to validate the TWAS associations. For this analysis, we clumped cis-eQTL for each feature (gene, transcript) using PLINK with 500 kb window and $r^2 > 0.1$. We selected the independent eQTLs clusters including top eQTL with

nominal $p$ values <0.05 and top PGC3 GWAS $p$ values < 0.05 for SMR analysis. For each cluster, we implemented SMR and HEIDI methods, using the SMR software tool (version 1.3.1) to examine the pleiotropic associations of genetic variation with feature expression in the RNA sequencing dataset and case–control status in the PGC3 GWAS schizophrenia data.

## Weighted gene coexpression network analysis (WGCNA)

We performed separate WGCNA in the samples of female ($N = 73$) and male placentae ($N = 74$) using transcript-level adjusted expression. WGCNA uses pairwise features correlation indices (biweight mid-correlation) as a continuous, i.e., weighted, measure of features relationships to detect sets of genes called modules[55]. To further obtain a robust network identification, we used the consensus network procedure. A consensus network is a compromise between two or more correlation matrices which attempt to obtain a more reliable estimate of pairwise correlations. Here, we applied a bootstrap fashion solution: ¾ of the sample was randomly extracted 100 times, so that 100 correlation matrices were generated. We used the blockwiseConsensusModules function in the WGCNA R package (version 1.69-81) in R (version 4.0) to detect modules using consensusQuantile = 0.2 as consensus option, which extracted the $2^{nd}$ decile values for each pairwise correlation along the 100 generated matrices to define the consensus matrix. This solution further decreased the impact of spurious large correlations[26,92].

We used a signed-hybrid network, in which negative correlations are set to zero (i.e., genes are considered unconnected) while positive correlation (i.e., connected genes) are transformed into an adjacency matrix by raising correlation coefficients to a positive exponent, $\beta$. The exponent $\beta$ is usually chosen to meet the scale-free power law connectivity distribution $p(k)$ by regressing $\log(p(k))$ on $\log(k)$. Scale invariance is widely considered a common organization feature of cellular functions. Higher $\beta$ values are more likely to generate a scale-free network, but also decrease the mean connectivity of the network. As a trade-off, it is usual practice to take the lowest $\beta$ value surviving the threshold of $R^2 > 0.8$ of the above-mentioned regression. The $\beta$ value was chosen using the *pickSoftThreshold* function in WGCNA R package. A hierarchical clustering method was then used to group genes into clusters, called modules, with a minimum module size of 40. Colors were used to arbitrarily label co-expression modules, with the gray module representing genes that did not cluster into any module. Co-expression was summarized by the Module Eigengene, i.e., the first principal component of the expression of genes in any given module[55,94].

## Stratified Linkage-Disequilibrium Score Regression (S-LDSC)

Following recommendations from the LDSC resource website (https://alkesgroup.broadinstitute.org/LDSCORE/)[109], S-LDSC was run for each list of variants (placenta eQTL, fetal brain eQTL, placenta&brain eQTL) with the baseline LD model v2.2 that included 97 annotations to control for the LD between variants with other functional annotations in the genome. We used HapMap Project Phase 3 SNPs as regression SNPs, and 1000 Genomes SNPs of European ancestry samples as reference SNPs, which were all downloaded from the LDSC resource website.

## Colocalization analyses

To perform colocalization analysis, we first implemented eQTL fine mapping by estimating priors from the MatrixQTL nominal results with the TORUS method[110]. Following the estimation of priors, we implemented DAP-G[67,111] to generate posterior inclusion probabilities that provide an estimate of the probability of a variant being causal for downstream colocalization with fastENLOC[68]. We applied fastENLOC within the schizophrenia GWAS PGC3 prioritized loci[10] that have index SNPs with GWAS $p < 1e-06$.

## Other statistical analyses

We analyzed the correlation between Z-scores from different TWAS analyses (e.g., in male and female samples, in placenta and brain, schizophrenia and other disorders/traits) using the *cor.test* function in R. A $\chi^2$-test with Yates correction was used to compare the overlap of TWAS significant genes in males and females in placenta and prenatal cortex brain, and to compare the overlap of TWAS significant genes in the placenta discovery datasets with the TWAS associations replicated ($p < 0.05$) with the same directionality (i.e., the sign of the Z-score) in the placenta replication dataset. We used a linear model in R to compare the number of TWAS hits across disorders, adjusting for the number of GWAS hits and GWAS sample size, and to compare the Z scores from different TWAS analyses. We also used linear models to compare the expression of schizophrenia TWAS genes across cell types, and between males and females in each cell type. We used the *geneSetTest* function on the moderated t-statistics from the differential expression analysis (performed in R, with the package limma[112], comparing placentae from SARS-CoV-2 positive and negative pregnancies) to test whether the schizophrenia TWAS genes tend to be more highly ranked in differential expression compared with randomly selected gene sets of the same size. We performed the same analysis using the TWAS genes associated with other disorders and traits, and immune gene sets from previous reference[113]. Low expressed genes in the placental and fetal brain RNA sequencing datasets and also in SARS-CoV-2 dataset were identified using the *expression_cutoff* function in the jaffelab R package[91], and excluded from this analysis. Bioturing BBrowser[114] (version 2.10.40, Bioturing Inc., San Diego, CA, USA; www.bioturing.com) was used to explore single cell data.

## Pathway, functional, and upstream regulator analyses

We investigated whether the placenta schizophrenia TWAS genes are enriched for particular biological features, by using QIAGEN's Ingenuity Pathway Analysis (IPA, QIAGEN, Redwood City, CA, USA; http://www.qiagen.com/ingenuity). The software determines the pathways, biological processes, and upstream regulator, enriched for a given set of genes by considering the number of focus genes that participate in each process and the total number of genes that are known to be associated with that process in the selected reference set. We performed the Ingenuity Pathway Analysis core analysis, using default parameters (reference set: Ingenuity Knowledge Base; relationships: direct and indirect; node types: all; data sources: all; confidence: experimentally observed and high; species: human, mouse, and rat; tissues and cell lines: all; mutations: all). We chose a P value calculation based on the Benjamini–Hochberg method of accounting for multiple testing in the canonical pathway and functional analyses. We also used IPA to perform a causal network[70] and an "Isoprofiler" analysis to identify respectively master regulators (Supplementary Data 55a) and diseases and functions (Supplementary Data 55b) associated with the 33 top-prioritized placenta- schizophrenia-specific risk, which show TWAS-significant association with schizophrenia in placenta, and not in fetal brain, are validated by SMR and colocalization analyses, and did not show any significant relevant association with any of the other disorders and traits analyzed.

## Ethics

Data from the RICHS cohort have been previously collected and the protocol of the RICHS study was reviewed and approved by the Office of Human Research Protection registered Institutional Review Boards at Women and Infants Hospital of Rhode Island and Emory University[88].

Prenatal cortical brain data were generated from tissue obtained from the UCLA Gene and Cell Therapy core according to IRB guidelines; this study[33] was performed under the auspices of the UCLA Office of Human Research Protection, which determined that it was exempt because samples are anonymous pathological specimens.

The placenta data of the replication dataset were generated in the context of a study[73] that has been approved by the Columbia University IRB Exp (protocol number IRB-AAAD9014).

## Reporting summary

Further information on research design is available in the Nature Portfolio Reporting Summary linked to this article.

## Data availability

All the data analyzed in this manuscript are publicly available and can be obtained via authorized access from the dbGaP database. Placenta RNA seq and genotype data are available on the NCBI SRA public database under accession code SRP095910 and on the dbGaP database under accession code phs001586.v1.p1 (discovery sample), and under accession code phs001717.v1.p1 (replication sample). Midgestational prenatal cortical brain RNAseq data are available on the dbGaP database under accession code phs001900.v1.p1.

## Code availability

This study did not generate new code. The scripts used for these analyses are available on request.

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

## Acknowledgements

We are grateful to the Lieber and Maltz families for their visionary support that funded the analytic work of this project. We are grateful to Richard Straub for their helpful review of the manuscript, to Nick Eagles for help on data processing, Sahil Patel for insights on gene functions, and to Laura Wortinger for helpful discussions. We thank all of the participants in the study and their families. We thank the Psychiatric Genomics Consortium for providing the GWAS summary statistics, and the authors of the transcriptomic studies to make these precious data publicly available. G.U., S.M., G.P., and D.R.W., received partial support from NIMH R21MH125108 (G.U., S.M., and G.P.), and from NICHD R01HD107140-01A1 (G.U., G.P., and D.R.W.).

Dataset(s) from the study "Genetic control of expression and splicing in developing human brain inform disease mechanisms" used for the analyses described in this manuscript were generated in the Geschwind laboratory and supported by NIH grants to Daniel H. Geschwind (5R37 MH060233, 5R01 MH094714, 1R01 MH110927) and to Jason L. Stein (R00MH102357). Dataset(s) were obtained from dbGaP found at http://www.ncbi.nlm.nih.gov/gap through dbGaP study accession numbers phs001900.v1.p1.

The TWAS replication analysis was performed using placenta RNAseq data available on dbGaP under accession phs001717.v1.p1. This research was supported by the Intramural Research Program of the Eunice Kennedy Shriver National Institute of Child Health and Human Development, National Institutes of Health (Contract Numbers: HHSN275200800013C; HHSN27500006; HHSN275200800003I;

HHSN275200800014C; HHSN275200800012C; HHSN2752008
00028C; HHSN275201000009C). For this study, we also want to
thank the Genomics Shared Resource of Roswell Park Cancer
Institute, supported by National Cancer Institute (NCI) grant
P30CA016056.

## Author contributions

G.U., P.D.C, and D.R.W. designed the study. G.U., G.P., and D.R.W.
interpreted the results. G.U., P.D.C, S.M., Q.C., S.H., J.K., and G.P.,
carried out statistical analyses. M.D., C.J.M., J.C., K.H., organized
subject recruitment and the collection of the placenta data, and
performed genotyping, and RNA sequencing. G.U., G.P., and D.R.W.
drafted the manuscript, and all authors contributed to the final
version of the paper.

## Competing interests

The authors declare no competing interests.

## Additional information

**Supplementary information** The online version contains
supplementary material available at

Gianluca Ursini or Daniel R. Weinberger.

**Peer review information** *Nature Communications* thanks Shefali Verma
and the other, anonymous, reviewer(s) for their contribution to the peer
review of this work. A peer review file is available.

