## [Peer Review File · Nature Communications]

Prioritization of potential causative genes for schizophrenia in placentaREVIEWER COMMENTS

Reviewer #1 (Remarks to the Author):

This manuscript is well written and describes a significant advancement in the genetic associations in schizophrenia. Through TWAS and SMR using placental and fetal brain tissues, authors identified many significant associations that 1) explain genetic mechanisms, 2) are sex-specific, and 3) correlate with other traits. The authors also explored the single-cell data to elucidate mechanisms related to different compartments and cell types of the placenta, which is unique. Lastly, since the pathways analyses identified a link with coronavirus pathogenesis pathways, an additional assessment of genomic risk between SCZ and maternal Sar-COV2 infections was performed. Authors use well-established methods and datasets to perform these analyses, and the details provided in the online methods sections are appropriate for the reproducibility of the results.

Major Concerns:

The association with mTOR signaling genes is interesting, as identified by the TWAS and pathway-based analyses. An important extension to understanding the role of the genes in this pathway would be to perform a Colocalization analysis to identify causal genes and variants that contributed to SCZ risk and gene expression in placental tissues. This approach applies to all analyses, whether sex-stratified, associations with other traits, etc. Therefore a complementary COLOC analysis is recommended.

TWAS studies are prone to high false positive rates. Therefore, a replication dataset or attempt to replicate the findings is necessary.

A large number of genes are identified as statistically significant in this study. An attempt at characterizing these genes that play a role in the brain and placenta and how those might influence the risk of SCZ would help orient the readers towards the bigger picture of this analysis.

Minor Concerns:

Figure 3i: It is unclear why the authors used the Bonferroni p-value for all results except fetal brain; they used FDR. Please explain why and I would suggest the use of the same Bonferroni. Is it that with the Bonferroni correction, no overlapping genes are present?

Figure 4 is hard to parse through. Clear distinctions on which quadrant reflects concordant and discordant associations should be labeled for clarity.

Reviewer #2 (Remarks to the Author):

In this manuscript, Ursini and colleagues describe interesting work identifying placental gene expression differences associated with genetic risk variants for schizophrenia. The methods, which include TWAS and SMR / HEIDI, appear to be performed correctly. My only comment is that, in the Introduction, as well as citing the currently unpublished work of Wortinger et al, the authors need to cite the recent study of Vassos et al (Schizophrenia Bulletin 2022) where an interaction between schizophrenia genomic risk and obstetric complications was not observed.

RESPONSE to REVIEWERS

Reviewer #1 (Remarks to the Author):

This manuscript is well written and describes a significant advancement in the genetic associations in schizophrenia. Through TWAS and SMR using placental and fetal brain tissues, authors identified many significant associations that 1) explain genetic mechanisms, 2) are sex-specific, and 3) correlate with other traits. The authors also explored the single-cell data to elucidate mechanisms related to different compartments and cell types of the placenta, which is unique. Lastly, since the pathways analyses identified a link with coronavirus pathogenesis pathways, an additional assessment of genomic risk between SCZ and maternal Sar-COV2 infections was performed. Authors use well-established methods and datasets to perform these analyses, and the details provided in the online methods sections are appropriate for the reproducibility of the results.

We are thankful to the reviewer for this comment and the overall appreciation of our work.

Major Concerns:

1.1. The association with mTOR signaling genes is interesting, as identified by the TWAS and pathway-based analyses. An important extension to understanding the role of the genes in this pathway would be to perform a Colocalization analysis to identify causal genes and variants that contributed to SCZ risk and gene expression in placental tissues. This approach applies to all analyses, whether sex-stratified, associations with other traits, etc. Therefore a complementary COLOC analysis is recommended.

We agree with the Reviewer that a Colocalization analysis (COLOC) represents a complementary approach to TWAS and SMR useful to prioritize not only candidate causal genes associated to diseases but also potential causative variants. We have performed a COLOC analysis^{1,2} in the whole sample and in subsets of female and male placentae. In addition to variant-level COLOC, we have performed a locus-level COLOC analysis. The latter approach has been recently shown to improve not only specificity but also sensitivity when using both TWAS and COLOC to detect biologically relevant genes with a possible causative role on disease². In the revised **Table S54**, we now report gene variant-level colocalization probability (GRCP, which is associated with regional colocalization probability, RCP) and gene locus-level colocalization probability (GLCP) for all the 502 genes whose expression in placenta at gene and/or transcript level is predicted to be associated with schizophrenia, in the whole sample, and/or in male, and/or in female placentae; we report detailed results from the COLOC analysis in **Tables S56-61**. Noteworthy genes from COLOC analyses are usually identified based on RCP/GRCP and GLCP values ≥ 0.50 ^{1,2}, or ≥ 0.10 ³. By using these criteria, we have identified 48 placental genes with GRCP/GLCP ≥ 0.50 , and 113 genes with GRCP/GLCP ≥ 0.10 . We found these results remarkable considering the lower number of schizophrenia risk genes that have been prioritized by COLOC analysis using transcriptomic datasets with larger sample size. For example, by using a similar approach, Benjamin and coll.⁴ (our colleagues at the Lieber Institute) prioritized 9 genes with regional colocalization probability (RCP) ≥ 0.50 in the caudate nucleus, leveraging transcriptomic data from 443 individuals; their analyses were performed in the whole sample, without stratifying by sex. In the latest schizophrenia GWAS published in *Nature* (2022), and earlier in *Cell* in 2020 (Pardinas), potential causative genes were highlighted with SMR, but colocalization analyses were not performed. In our datasets of 147 placentae, we find 38 genes with RCP ≥ 0.50 in the whole sample, without stratifying by sex; as predictable, these genes have also GRCP ≥ 0.50 , given that GRCP values summarize RCP values. By adding the analyses by sex, we identify a total of 42 genes (4 more genes) with RCP and GRCP ≥ 0.50 , and 48 placental genes with

RCP/GRCP or GLCP ≥ 0.50 . We believe that these results support the hypothesis of a relevance of the placental transcriptome in mediating part of the genomic risk for schizophrenia. In the revised manuscript, we are using the COLOC results to identify - among the 139 prioritized genes - 33 top-prioritized genes, i.e. further validated by COLOC. Of note, the schizophrenia risk genes prioritized by COLOC include genes associated with the mTOR pathway, such as RPS17⁵, ESAM⁶ and ESAM-AS⁶, CENPM⁷, NRG1⁸, CD14⁹, DGCR8¹⁰⁻¹², RNF38¹³, VSIG2 (a gene involved in protein-protein interaction with mTOR: <https://thebiogrid.org/interaction/3096026/vsig2-mtor.html#>). Moreover, a causal network analysis¹⁴ in IPA shows that two top master regulators of 13 and 14 of these 33 top-prioritized genes are respectively WNT7 (p= 0.0000109, $p_{B-Hcorrected}=0.0129$) and IL-2R (p= 0.000017, $p_{B-Hcorrected}=0.0129$), which are known activators of the mTOR pathway^{15,16} (**Table S55**). We thank the reviewer for this suggestion, which provides a further criterion for the prioritization of placental genes potentially causative for schizophrenia. At the end of the results section, we have now added a paragraph summarizing these results ("**Colocalization and replication analyses support the relationship between the placenta transcriptome and genomic risk for schizophrenia**")

1.2. TWAS studies are prone to high false positive rates. Therefore, a replication dataset or attempt to replicate the findings is necessary.

We agree with the reviewer about the importance of replication and, indeed, we are going to generate in our lab well-powered placental transcriptomic datasets that will be used also for this purpose. Unfortunately, these datasets will not be available in the short term; therefore, we have explored the possibility of analyzing other publicly available placental transcriptomic datasets. We have contacted the corresponding authors of a paper focused on the analyses of placenta transcriptomic data¹⁷. Unfortunately, genotype data are not yet available, so that a replication analysis is not possible in this well-powered sample. We have instead obtained dbGaP access (Study Accession: phs001717.v1.p1) to transcriptomic and genomic data from 70 placentae from the Eunice Kennedy Shriver National Institute of Child Health and Human Development (NICHD) Fetal Growth Study¹⁸. Unfortunately, the related raw sequencing data were not publicly available, so that we could not use our RNAseq processing pipeline to obtain expression data at gene and transcript level. We therefore used only the gene expression data available as FPKM on the dbGaP website. In the whole dataset (N=70), we detected 4850 genes with heritable gene expression, 3535 of which overlapped the 8558 genes with heritable cis-regulated expression in our discovery sample. 57 of these 3535 genes were among the 187 TWAS-significant genes in the analyses in our whole discovery sample. The TWAS association was replicated (p<0.05) for 21 out of these 57 genes with the same directionality (i.e., Z-score sign), thus showing a level of replication higher than expected by chance (χ^2 with Yates correction = 29.79, p<0.0001). Consistently, we found a significant positive (**Fig.S3a**; Fig.S3 is also reported below for the reviewer's convenience) and concordant (**Fig.S3b**) association between the Z scores from the two TWAS analyses in the discovery and in the replication samples (relationship between Z scores: t= 23.40, r=0.36, Rho=0.40, p<2e-16; **Fig. S3a,b**). The correlation was stronger when considering only TWAS-significant genes in the discovery sample (r=0.51, Rho=0.60, p=5.02e-05; **Fig.S3a**).

In the female dataset (N=35), we detected 5363 genes with heritable gene expression, 3856 of which overlapped the genes with heritable cis-regulated expression in our discovery sample. 49 out of these 3856 genes were among the 225 TWAS-significant unique genes in the analyses in our female discovery sample. The TWAS association was replicated (p<0.05) for 13 out of these 49 genes with the same directionality (i.e., Z-score sign), thus showing a level of replication higher than expected by chance (χ^2 with Yates correction = 7.82, p=0.005). Consistently, we found a significant positive (**Fig.S3c**) and concordant (**Fig.S3d**) association between the Z scores from the two TWAS analyses in the discovery and

in the replication samples (relationship between Z scores: $t = 22.24$, $r = 0.34$, $Rho = 0.39$, $p < 2e-16$; **Fig. S3c,d**). The correlation was slightly stronger when considering only TWAS-significant genes in the discovery sample ($r = 0.40$, $Rho = 0.42$, $p = 0.0025$; **Fig. S3c**).

In the male dataset ($N = 35$), we detected 5299 genes with heritable gene expression, 3764 of which overlapped the genes with heritable cis-regulated expression in our discovery sample. 43 out of these 3764 genes were among the 215 TWAS-significant unique genes in the analyses in our male discovery sample. The TWAS association was replicated ($p < 0.05$) for 22 out of these 43 genes with the same directionality (i.e., Z-score sign), thus showing a level of replication higher than expected by chance (χ^2 with Yates correction = 61.02, $p < 0.0001$). Consistently, we found a significant positive (**Fig. S3e**) and concordant (**Fig. S3f**) association between the Z scores from the two TWAS analyses in the discovery and in the replication samples (relationship between Z scores: $t = 24.24$, $r = 0.37$, $Rho = 0.40$, $p < 2e-16$; **Fig. S3e,f**). The correlation was stronger when considering only TWAS-significant genes in the discovery sample ($r = 0.73$, $Rho = 0.65$, $p = 3.2e-08$; **Fig. S3e**).

Given the limited sample size of this dataset, a more cautious approach would be not to draw strong conclusions from these results; we would prefer, therefore, not to use these results to further prioritize placental genes with a potential causative role on risk for schizophrenia. However, considered as a whole, these analyses support the reliability of our findings and the relevance of the placental transcriptome in mediating the effect of genomic risk for schizophrenia. At the end of the results section, we have now added a paragraph summarizing these results (“*Colocalization and replication analyses support the relationship between the placenta transcriptome and genomic risk for schizophrenia*”). Replication analyses are also presented in a **Supplementary Note** (“TWAS replication”). **Tables S62-65** report statistics from the replication analyses.

Fig. S3. TWAS replication in placenta. Relationship between the Z-scores from the schizophrenia TWAS analyses, performed in the whole sample (**A,B**), in the female (**C,D**) and in the male sample (**E,F**). Z-scores are from the TWAS performed in the discovery placental dataset¹⁹ ($N = 147$: 73 females and 74 males) and in the replication placental dataset¹⁸ ($N = 70$: 35 females and 35 males). **A,C,E**: Scatterplots of the correlation between the Z scores from the TWAS analyses in the discovery dataset (x-axis) and in the replication dataset (y-axis), in the whole sample (**A**), and in placentae from female (**C**), and male (**E**) offspring. Genes TWAS-significant in the discovery dataset are highlighted as red dots. Pearson correlation coefficients (“r”) are reported for the

total number of genes (black font) and for the TWAS-significant genes in the discovery dataset (red font). **B,D,F**: RRHO²⁰ heatmaps showing concordant (bottom-left and top-right quadrant in each panel, highlighted in a red frame) and discordant (top-left and bottom-right quadrant in each panel, highlighted in a green frame) TWAS association with schizophrenia in the discovery and in the replication dataset, in the whole sample (**B**), and in placentae from female (**D**) and male (**F**) offspring. Concordant and discordant associations are estimated using a threshold-free algorithm based on a rank-rank hypergeometric overlap approach, as described elsewhere²⁰. Color bar represents negative logarithm of p-value of the overlap.

1.3. *A large number of genes are identified as statistically significant in this study. An attempt at characterizing these genes that play a role in the brain and placenta and how those might influence the risk of SCZ would help orient the readers towards the bigger picture of this analysis.*

We agree with the reviewer that a functional characterization of these genes would be crucial to put them in context. We are now providing additional information on all the top-prioritized and selected placental genes with a potentially causative role for schizophrenia in the revised **Table S54**, in **Table S55**, and in the **Supplementary Note “Selected placental genes potentially causative for schizophrenia”**, to facilitate the reader to link information about genes with possible mechanisms of risk mentioned in the discussion. We indeed believe that this was a missing information in the previous version of the manuscript, and we thank the reviewer for their suggestion.

Minor Concerns:

1.4. *Figure 3i: It is unclear why the authors used the Bonferroni p-value for all results except fetal brain; they used FDR. Please explain why and I would suggest the use of the same Bonferroni. Is it that with the Bonferroni correction, no overlapping genes are present?*

We apologize for having created some confusion in this regard. We would like to reassure the reviewer that the overlapping genes between placenta and fetal brain were identified using the same Bonferroni threshold (<0.05) in placenta and in fetal brain (**Fig.3j in the manuscript**). In fetal brain, we have used an FDR threshold only to have a conservative criterion to identify placenta-specific TWAS- genes, that is, genes that are TWAS significant in placenta with Bonferroni p-value <0.05 , and are not TWAS significant in brain, having in brain not only a Bonferroni p-value >0.05 , but also a FDR corrected p-value >0.01 (**Fig.3i in the manuscript**). We think that this is a conservative criterion to decrease the risk of considering, as placenta-specific TWAS-genes, false negative TWAS-genes in fetal brain. Indeed, with this criterion (Bonferroni corrected p-value in placenta <0.05 and FDR-corrected p-value in fetal brain >0.01) we identify 206 placenta-specific TWAS, while the placenta-specific TWAS genes would be 303 using Bonferroni corrected p-value in placenta <0.05 and Bonferroni-corrected p-value in fetal brain >0.05 . Because one aim of our work is to prioritize potentially causative genes for schizophrenia which are more likely to play an etiological role in placenta, and not in fetal brain, we prefer to keep the current criterion to define placenta-specificity. In the current version of the manuscript, we clarify the rationale for this criterion to define placenta-specificity, we specify that overlapping genes were detected using Bonferroni-corrected p-values in both organs, and we report also results related to placenta specificity using only Bonferroni-corrected p-values.

1.5. *Figure 4 is hard to parse through. Clear distinctions on which quadrant reflects concordant and discordant associations should be labeled for clarity.*

We thank the reviewer for this recommendation. We have highlighted the distinction between quadrants reflecting concordant and discordant associations, by putting them in a red and green frame respectively. We'll forward this recommendation to the curators of the RRHO2 package that we have used to create these heatmaps.

Reviewer #2 (Remarks to the Author):

In this manuscript, Ursini and colleagues describe interesting work identifying placental gene expression differences associated with genetic risk variants for schizophrenia. The methods, which include TWAS and SMR / HEIDI, appear to be performed correctly.

We thank the reviewer for this comment and the appreciation of our work.

2.1. My only comment is that, in the Introduction, as well as citing the currently unpublished work of Wortinger et al, the authors need to cite the recent study of Vassos et al (Schizophrenia Bulletin 2022) where an interaction between schizophrenia genomic risk and obstetric complications was not observed.

In the revised manuscript we cite, in addition to the preprint from Wortinger et al²¹, also the study from Vassos et al.²², the two commentaries on this article^{23,24}, and a further article²⁵ that announce on-going research on this topic, as follows:

“One subsequent study²² that used a less detailed inventory for the assessment of ELCs and that did not survey many of the ELCs associated with schizophrenia^{26,27} did not observe a statistical interaction between genomic risk and ELCs²². In contrast, the principal findings from our two studies^{23,24} have been largely replicated in a recent study from Norway²¹, which detected an interaction between placental genomic risk for schizophrenia and serious ELCs associated with birth asphyxia. These results have triggered broader perspectives in gene by environment research related to psychiatric illness²⁵ and...”

References:

- 1 Wen, X., Pique-Regi, R. & Luca, F. Integrating molecular QTL data into genome-wide genetic association analysis: Probabilistic assessment of enrichment and colocalization. *PLoS Genet* **13**, e1006646, doi:10.1371/journal.pgen.1006646 (2017).
- 2 Hukku, A., Sampson, M. G., Luca, F., Pique-Regi, R. & Wen, X. Analyzing and reconciling colocalization and transcriptome-wide association studies from the perspective of inferential reproducibility. *Am J Hum Genet* **109**, 825-837, doi:10.1016/j.ajhg.2022.04.005 (2022).
- 3 Al-Barghouthi, B. M. *et al.* Transcriptome-wide association study and eQTL colocalization identify potentially causal genes responsible for human bone mineral density GWAS associations. *Elife* **11**, doi:10.7554/eLife.77285 (2022).
- 4 Benjamin, K. J. M. *et al.* Analysis of the caudate nucleus transcriptome in individuals with schizophrenia highlights effects of antipsychotics and new risk genes. *Nat Neurosci* **25**, 1559-1568, doi:10.1038/s41593-022-01182-7 (2022).
- 5 Jiang, X., Feng, S., Chen, Y., Feng, Y. & Deng, H. Proteomic analysis of mTOR inhibition-mediated phosphorylation changes in ribosomal proteins and eukaryotic translation initiation factors. *Protein Cell* **7**, 533-537, doi:10.1007/s13238-016-0279-0 (2016).
- 6 Diener, N. *et al.* Posttranslational modifications by ADAM10 shape myeloid antigen-presenting cell homeostasis in the splenic marginal zone. *Proc Natl Acad Sci U S A* **118**, doi:10.1073/pnas.2111234118 (2021).
- 7 Liu, C. *et al.* Upregulation of CENPM facilitates lung adenocarcinoma progression via PI3K/AKT/mTOR signaling pathway. *Acta Biochim Biophys Sin (Shanghai)* **54**, 99-112, doi:10.3724/abbs.2021013 (2022).
- 8 Zhou, Y. *et al.* Interactome analysis reveals ZNF804A, a schizophrenia risk gene, as a novel component of protein translational machinery critical for embryonic neurodevelopment. *Mol Psychiatry* **23**, 952-962, doi:10.1038/mp.2017.166 (2018).
- 9 Li, Y. *et al.* M860, a Monoclonal Antibody against Human Lactoferrin, Enhances Tumoricidal Activity of Low Dosage Lactoferrin via Granzyme B Induction. *Molecules* **24**, doi:10.3390/molecules24203640 (2019).
- 10 Sun, Y. *et al.* N(6)-methyladenosine-dependent pri-miR-17-92 maturation suppresses PTEN/TMEM127 and promotes sensitivity to everolimus in gastric cancer. *Cell Death Dis* **11**, 836, doi:10.1038/s41419-020-03049-w (2020).
- 11 Nallamshetty, S., Chan, S. Y. & Loscalzo, J. Hypoxia: a master regulator of microRNA biogenesis and activity. *Free Radic Biol Med* **64**, 20-30, doi:10.1016/j.freeradbiomed.2013.05.022 (2013).
- 12 Gong, W. J., Li, R., Dai, Q. Q. & Yu, P. METTL3 contributes to slow transit constipation by regulating miR-30b-5p/PIK3R2/Akt/mTOR signaling cascade through DGCR8. *J Gastroenterol Hepatol* **37**, 2229-2242, doi:10.1111/jgh.15994 (2022).
- 13 Zhou, J., Tang, Z. Y. & Sun, X. L. RNF38 inhibits osteosarcoma cell proliferation by binding to CRY1. *Biochem Cell Biol* **99**, 629-635, doi:10.1139/bcb-2021-0093 (2021).
- 14 Kramer, A., Green, J., Pollard, J., Jr. & Tugendreich, S. Causal analysis approaches in Ingenuity Pathway Analysis. *Bioinformatics* **30**, 523-530, doi:10.1093/bioinformatics/btt703 (2014).

- 15 von Maltzahn, J., Bentzinger, C. F. & Rudnicki, M. A. Wnt7a-Fzd7 signalling directly activates the Akt/mTOR anabolic growth pathway in skeletal muscle. *Nat Cell Biol* **14**, 186-191, doi:10.1038/ncb2404 (2011).
- 16 Ray, J. P. *et al.* The Interleukin-2-mTORc1 Kinase Axis Defines the Signaling, Differentiation, and Metabolism of T Helper 1 and Follicular B Helper T Cells. *Immunity* **43**, 690-702, doi:10.1016/j.immuni.2015.08.017 (2015).
- 17 Gong, S. *et al.* The RNA landscape of the human placenta in health and disease. *Nat Commun* **12**, 2639, doi:10.1038/s41467-021-22695-y (2021).
- 18 Delahaye, F. *et al.* Genetic variants influence on the placenta regulatory landscape. *PLoS Genet* **14**, e1007785, doi:10.1371/journal.pgen.1007785 (2018).
- 19 Paquette, A. G. *et al.* Placental FKBP5 genetic and epigenetic variation is associated with infant neurobehavioral outcomes in the RICHS cohort. *PLoS One* **9**, e104913, doi:10.1371/journal.pone.0104913 (2014).
- 20 Cahill, K. M., Huo, Z., Tseng, G. C., Logan, R. W. & Seney, M. L. Improved identification of concordant and discordant gene expression signatures using an updated rank-rank hypergeometric overlap approach. *Sci Rep* **8**, 9588, doi:10.1038/s41598-018-27903-2 (2018).
- 21 Wortinger L, S. A., Szabo A, Nerland S, Smelror R, Jørgensen K, Barth C, Andreou D, Thoresen M, Andreassen O, Djurovic S, Ursini G, Agartz I. . The impact of placental genetic risk for schizophrenia and birth asphyxia on brain development. *PREPRINT (Version 1) available at Research Square*, doi:<https://doi.org/10.21203/rs.3.rs-1626382/v1> (2022).
- 22 Vassos, E. *et al.* Lack of Support for the Genes by Early Environment Interaction Hypothesis in the Pathogenesis of Schizophrenia. *Schizophr Bull* **48**, 20-26, doi:10.1093/schbul/sbab052 (2022).
- 23 Ursini, G. *et al.* Convergence of placenta biology and genetic risk for schizophrenia. *Nat Med* **24**, 792-801, doi:10.1038/s41591-018-0021-y (2018).
- 24 Ursini, G. *et al.* Placental genomic risk scores and early neurodevelopmental outcomes. *Proc Natl Acad Sci U S A* **118**, doi:10.1073/pnas.2019789118 (2021).
- 25 Braun, A., Kraft, J. & Ripke, S. Study protocol of the Berlin Research Initiative for Diagnostics, Genetics and Environmental Factors in Schizophrenia (BRIDGE-S). *BMC Psychiatry* **23**, 31, doi:10.1186/s12888-022-04447-4 (2023).
- 26 Ursini, G. & Weinberger, D. R. Replicating G x E: The Devil and the Details. *Schizophr Bull* **48**, 4, doi:10.1093/schbul/sbab109 (2022).
- 27 Vassos, E. & Murray, R. M. The Jury Is Still out on Placental Genes and Obstetric Complications. *Schizophr Bull* **48**, 5, doi:10.1093/schbul/sbab117 (2022).

REVIEWERS' COMMENTS

Reviewer #1 (Remarks to the Author):

The authors have revised the manuscript appropriately. I have no more concerns.

Reviewer #2 (Remarks to the Author):

The authors have addressed my comment

RESPONSE to REVIEWERS

Reviewer #1 (Remarks to the Author):

The authors have revised the manuscript appropriately. I have no more concerns.

Reviewer #2 (Remarks to the Author):

The authors have addressed my comment

We thank the reviewers for their precious advice, previously provided, and we are glad for their comments, which reflect an appreciation of our work.